# Directed evolution of broadly crossreactive chemokine-blocking antibodies efficacious in arthritis

Alessandro Angelini [1,2,3], Yoshishige Miyabe [4], Daniel Newsted[1], Byron H. Kwan[1,5], Chie Miyabe[4], Ryan L. Kelly [1,5], Misha N. Jamy[1], Andrew D. Luster [4] & K. Dane Wittrup[1,2,5]

Chemokine receptors typically have multiple ligands. Consequently, treatment with a blocking antibody against a single chemokine is expected to be insufficient for efficacy. Here we show single-chain antibodies can be engineered for broad crossreactivity toward multiple human and mouse proinflammatory ELR$^+$ CXC chemokines. The engineered molecules recognize functional epitopes of ELR$^+$ CXC chemokines and inhibit neutrophil activation ex vivo. Furthermore, an albumin fusion of the most crossreactive single-chain antibody prevents and reverses inflammation in the K/BxN mouse model of arthritis. Thus, we report an approach for the molecular evolution and selection of broadly crossreactive antibodies towards a family of structurally related, yet sequence-diverse protein targets, with general implications for the development of novel therapeutics.

[1] Koch Institute for Integrative Cancer Research, Massachusetts Institute of Technology, 500 Main Street, Cambridge, MA 02139, USA. [2] Department of Chemical Engineering, Massachusetts Institute of Technology, 77 Massachusetts Avenue, Cambridge, MA 02139, USA. [3] Department of Molecular Sciences and Nanosystems, Ca' Foscari University of Venice, Via Torino 155, Venezia Mestre 30172, Italy. [4] Center for Immunology and Inflammatory Diseases, Division of Rheumatology, Allergy and Immunology, Massachusetts General Hospital, Harvard Medical School, 149 Thirteenth Street, Charlestown, MA 02129, USA. [5] Department of Biological Engineering, Massachusetts Institute of Technology, 77 Massachusetts Avenue, Cambridge, MA 02139, USA. Correspondence and requests for materials should be addressed to A.A. (email: alessandro.angelini@unive.it) or to K.D.W. (email: wittrup@mit.edu)

Chronic inflammatory diseases usually involve multiple ligands that act synergistically through promiscuous and diverse receptors[1]. This complexity is exemplified by the ELR$^+$ CXC chemokine system, a large family of secreted proteins that have a prominent function in the development and progression of numerous inflammatory diseases, including rheumatoid arthritis (RA)[2]. The ELR$^+$ CXC chemokines are so named because of the presence of an amino terminal Glu-Leu-Arg (ELR) amino-acid motif followed by two invariant cysteines (C) that are separated by a random residue (X)[3]. The ELR$^+$ CXC chemokine system includes many small and structurally similar chemoattractant ligands capable of binding to and activating the related CXCR1 and CXCR2 G protein-coupled receptors (GCPR) expressed on the surface of neutrophils[4]. These ligands function either by autocrine or paracrine mechanisms to induce signaling networks that direct neutrophils to sites of inflammation. Importantly, increased levels of ELR$^+$ CXC chemokines have been detected in the sera, synovial fluid, and synovial tissue of patients with RA[5–8]. Studies in animal models have demonstrated that genetic deletion of the most promiscuous ELR$^+$ CXC chemokine receptor, CXCR2, can block the development of joint inflammation in anti-type II collagen antibody-induced arthritis[9],

adjuvant-induced arthritis[10–12], and K/BxN serum transfer-induced arthritis[13,14]. This evidence indicates that the ELR$^+$ CXC chemokine signaling network is an attractive therapeutic target for the treatment of arthritic diseases[15].

Inhibition of ELR$^+$ CXC chemokine-driven signaling has been attempted with various antagonists against CXCR1 and CXCR2 receptors, including neutralizing antibodies, small molecules, and peptide-derived inhibitors. Despite the broad variety of approaches, these conventional receptor-based therapies have mostly had limited success in the clinic[16,17]. Failures have often been attributed to (i) differences between the orthologous rodent (preclinical) and human (clinical) systems and (ii) the extremely high doses of antagonist required to guarantee continuous receptor occupancy, such that all receptors in the body are antagonized[16,17]. The latter phenomenon is particularly problematic, as CXCR1 and CXCR2 undergo rapid internalization cycles (with a half-life of 6–8 h) and are expressed on neutrophils, which are the most abundant (40–75%) of the circulating leukocytes and have short circulating half-life (6–8 h)[18]. As a result, antagonists that target these receptors are cleared quickly, reducing the amount of drug available in circulation. Efforts to develop more effective receptor-based therapies have led to the discovery of non-

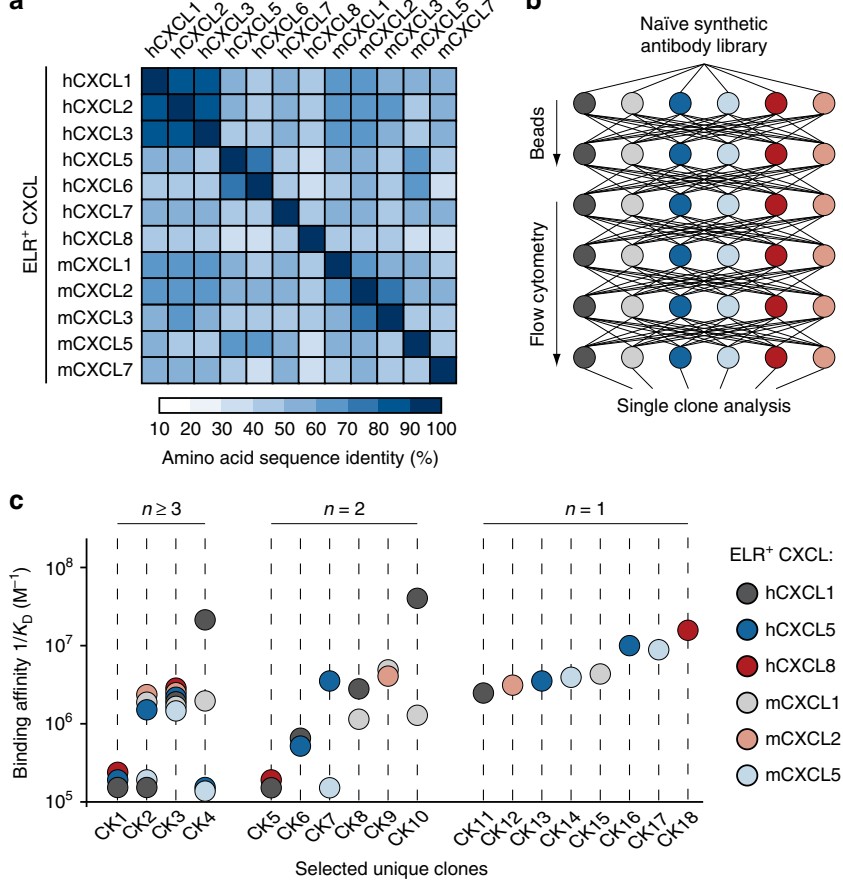

**Fig. 1** Isolation of crossreactive antibodies toward multiple ELR$^+$ CXC chemokines. **a** Heat map displaying the sequence identity among multiple human and murine ELR$^+$ CXC chemokines. The color of each element in the heat map indicates the sequence identity percentage, ranging from 10% (white) to 100% (dark blue). "h" and "m" indicates human and murine CXC chemokines, respectively. **b** Schematic representation of the iterative selection pathways applied to isolate crossreactive molecules from a naïve library of synthetic antibodies displayed on the surface of yeast. Two cycles of magnetic bead screening followed by four cycles of flow cytometry sorting were applied. **c** Plot of the binding affinities of 18 unique yeast-displayed synthetic antibody protein binders (CK) selected using six diverse human (hCXCL1, hCXCL5, and hCXCL8) and murine (mCXCL1, mCXCL2, and mCXCL5) ELR$^+$ CXC chemokine ligands. Each chemokine and its corresponding binding affinity values are reported as differently colored filled circles and indicate the means of at least three independent experiments. Data are presented as inverse equilibrium binding constants (1/$K_D$; M$^{-1}$)

competitive allosteric modulators of CXCR1 and CXCR2[19,20]. This class of inhibitors seems to provide unique advantages over conventional drug formats and is being tested in advanced clinical trials[21].

One alternative approach to inhibit CXCR1 and CXCR2 signaling is the blockade of the ELR[+] CXC chemokine ligands, which are often spatially confined to precise anatomical locations and might enable improved drug accumulation and specificity. However, generating synthetic compounds that antagonize these ligands has proven difficult, owing to their small size and lack of molecular pits or grooves. Many monoclonal antibodies targeting single ELR[+] CXC chemokines with high affinity and specificity have been developed, but, despite their

potency and low toxicity, single neutralizing antibody-based therapies have failed to block disease progression[22–27]. This limited therapeutic efficacy is often attributed to the multifactorial and redundant nature of the ELR[+] CXC chemokine system. Consistent with this hypothesis, therapeutic intervention using a cocktail of two or three monoclonal antibodies has resulted in synergistic potency, suggesting that augmented efficacy might be achieved by neutralizing multiple ligands at once[24,28].

In the present study, we use yeast-display technology to engineer serum albumin (SA)–antibody fusions that can simultaneously block multiple orthologous human and mouse ligands, thus providing the advantages of broad neutralization within a single molecule. Importantly, these fusions demonstrate

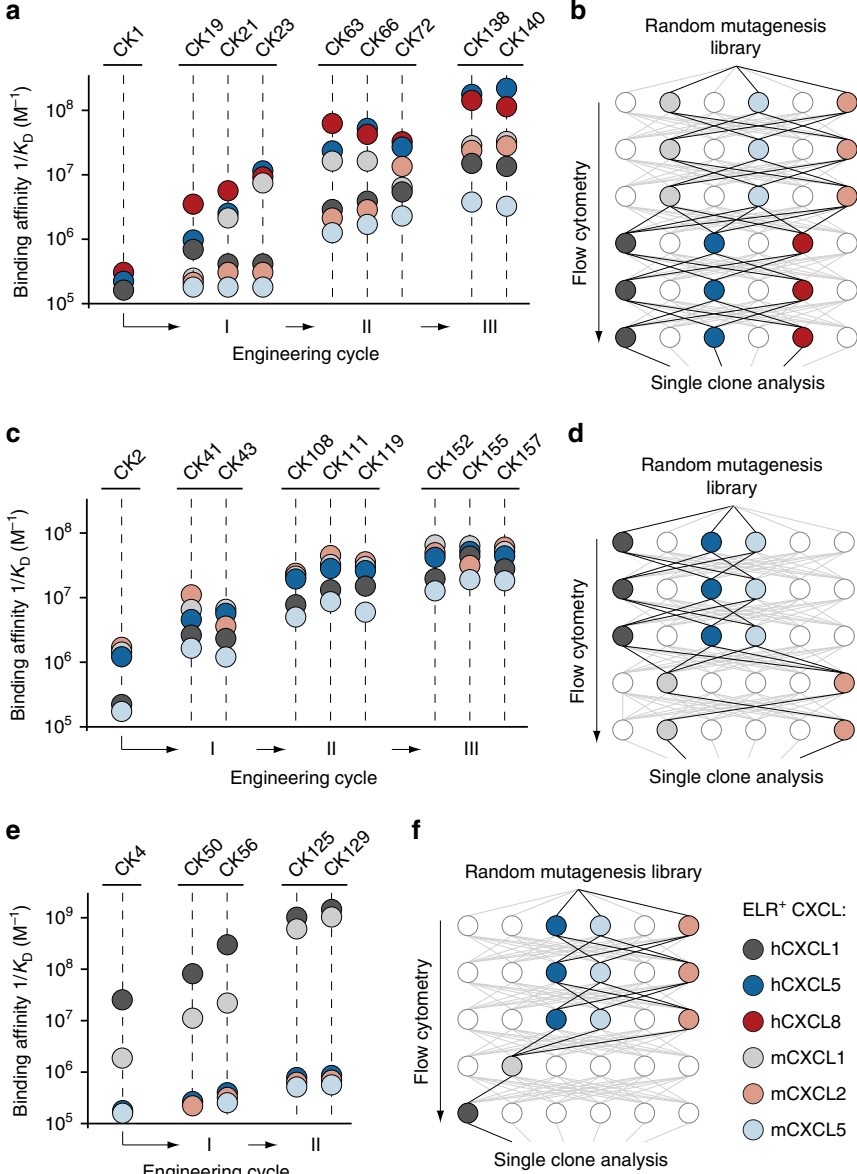

**Fig. 2** Molecular co-evolution of antibody affinity and crossreactivity. Plots displaying binding affinities of engineered clones derived from **a** CK1, **c** CK2, and **e** CK4 lineages. Two independent processes of selection (I and II), each including the generation of random yeast-display antibody libraries and six cycles of flow cytometry sorting, followed by a third round of site-directed mutagenesis (III), were performed. ELR[+] CXC chemokines and their corresponding binding affinity values are reported as differently colored filled circles and indicate the means of at least three independent experiments. Data are presented as inverse equilibrium binding constants ($1/K_D$; M[−1]). Selection pathways applied to isolate crossreactive molecules from a mutagenized yeast-display synthetic antibody library that yielded **b** CK1-, **d** CK2-, and **f** CK4-derived clones with improved binding affinity and crossreactivity. Each pathway comprises five to six cycles of flow cytometry sorting

promising prophylactic and therapeutic efficacy in vivo in the K/BxN mouse model of inflammatory arthritis. Thus, we show that directed evolution could be used to develop next-generation therapeutics against multiple redundant pathological factors.

## Results

**Isolation of crossreactive antibodies.** To evolve highly crossreactive protein binders toward multiple proinflammatory ELR[+]

CXC chemokines, we used synthetic single-chain variable antibody fragment libraries displayed on the surface of yeast[29,30]. For selection purposes, we chose to target three human (hCXCL1, hCXCL5, and hCXCL8) and three murine (mCXCL1, mCXCL2, and mCXCL5) chemokines based on their low sequence identity (Fig. 1a and Supplementary Table 1) and proven therapeutic relevance[8,14]. We produced the selected chemokines and confirmed their purity as well as their biological activity based on their ability to mobilize intracellular calcium in primary

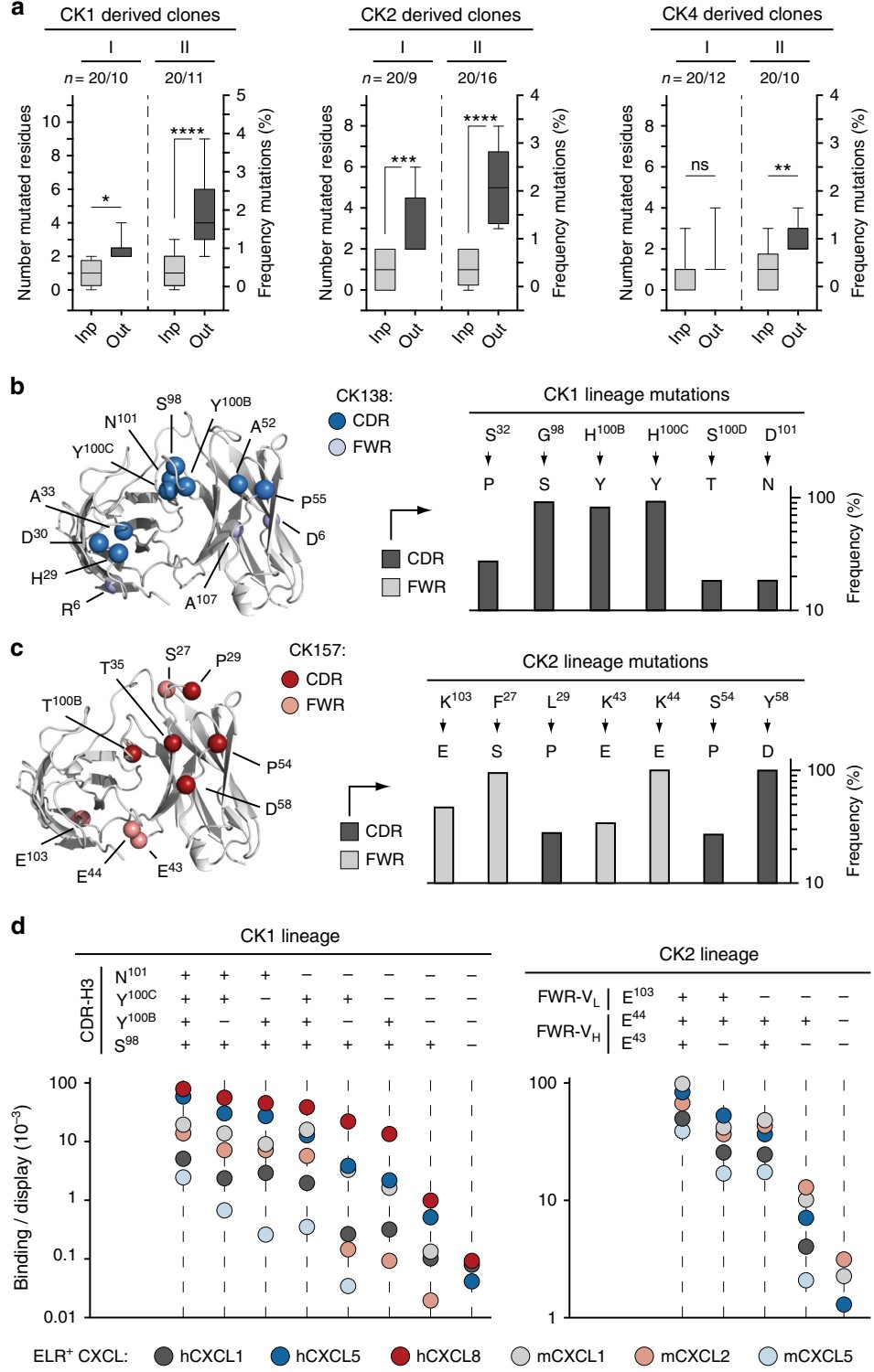

neutrophils (Supplementary Fig. 1a–d). To favor the selection of crossreactive antibodies, we implemented a combinatorial approach in which the output of each selection cycle was exposed to a diverse array of ELR$^+$ CXC chemokines in the next cycle (Fig. 1b). We took advantage of highly avid reagents preloaded with ELR$^+$ CXC chemokines to ensure that weak crossreactive antibodies were also isolated. Subsequent DNA sequencing of selected clones revealed the presence of 18 unique antibody fragments with varying amino-acid compositions and loop lengths within the complementarity-determining regions (CDRs). Selected antibodies exhibited diverse affinities and specificities for soluble ELR$^+$ CXC chemokines (Fig. 1c, Supplementary Fig. 2, and Supplementary Table 2). When monomeric chemokines were used to assess binding, only four of these clones (CK1–CK4) recognized at least three different ELR$^+$ CXC chemokines. The most abundant and crossreactive of these antibodies (CK3) recognized the biotinylation sequence located at the C-terminus of each chemokine, thus explaining its exquisite promiscuity and similar binding affinities for the various chemokines. However, CK1, CK2, and CK4 remained as promising crossreactive candidates for further development. The remaining 14 antibody clones were either bi-specific (CK5–CK10) or mono-specific (CK11–CK18). Overall, this selection strategy yielded three crossreactive antibodies that occurred at lower frequency within the total selected variant pool and had weaker binding affinities compared with the mono- and bi-specific antibodies.

**Molecular co-evolution of affinity and crossreactivity**. In order to further improve both the binding affinity and crossreactivity of CK1, CK2, and CK4 clones, we deviated from our previous selection strategy, which focused only on crossreactivity, and implemented a novel selection approach to co-evolve affinity and crossreactivity simultaneously. We first applied error-prone PCR amplification to introduce genetic diversity into our antibody-encoding genes. We then promoted the selection of clones displaying increased binding affinity by allowing the mutants to evolve through sequential cycles of equilibrium-based selection using decreasing concentrations of ELR$^+$ CXC chemokines. Concomitantly, we forced the development of crossreactivity by exposing the outputs of each affinity selection cycle towards a different ELR$^+$ CXC chemokine in the following cycle. During this iterative process, we solely collected variants whose affinity and crossreactivity towards ELR$^+$ CXC chemokines was higher than that of their respective parental clones. After two iterative evolutionary processes, each comprising five to six consecutive cycles of selection, we sequenced the isolated clones and assessed their binding affinity and crossreactivity towards ELR$^+$ CXC chemokines (Supplementary Figs. 3–5 and Supplementary

Table 3). Additionally, for cases in which we found distinct affinity-conferring mutations scattered across different clones, we combined mutations together to investigate the possibility of even further promiscuity and higher affinity. A summary of the overall co-evolutionary approach, including two iterative selection processes for crossreactivity and affinity (I and II) and a third cycle of combinatorial mutagenesis (III), is shown in Fig. 2a, c, e.

This two-pressure selection strategy yielded antibodies with improved affinity and, in most cases, increased crossreactivity toward multiple ELR$^+$ CXC chemokines. For example, the engineered CK138 clone recognized double the number of chemokines (from three to six) and achieved roughly a 30- to 340-fold improvement in affinity toward these chemokines ($K_D$ values ranging from 5.8 to 193 nM) relative to the parental CK1 clone. Similarly, the CK157 clone displayed crossreactivity toward five targets and added a 20- to 55-fold improvement in affinity ($K_D$ values ranging from 16.9 to 57.1 nM) as compared to the initial CK2 clone. Finally, while CK129 retained only minimal crossreactivity toward two targets, we observed a significant increase in affinity of 50- and 800-fold, respectively, toward human hCXCL1 ($K_D = 0.79$ nM) and its mouse homolog mCXCL1 ($K_D = 0.93$ nM).

Importantly, the sequential order in which the ELR$^+$ CXC chemokine targets were exposed to the antibody mutant libraries was critical to the success of the selection process. Among all the possible selection pathways, we observed improvements in both affinity and crossreactivity only when recombinant genetic libraries were screened in order from lowest to highest affinity chemokines (Fig. 2b, d). Note that this was not applicable to the development of CK129, as its parental clone (CK4) already possessed high initial affinity toward hCXCL1 and mCXCL1, but negligible affinity towards the other chemokines (Fig. 2f). Overall, our two-pressure selection approach promoted the evolution of crossreactive molecules with improved affinity and revealed the importance of the selection pathway for the achievement of crossreactivity.

**Crossreactive antibodies have a high level of mutations**. Although we applied reaction conditions that allowed, on average, one to two amino-acid mutations per gene, selected clones from each round of sorting showed higher mutation rates (Fig. 3a). While the crossreactive antibody CK138 predominantly gathered mutations within the CDRs, CK157 collected numerous mutations within the framework regions (FWRs) (Fig. 3b, c and Supplementary Fig. 6). Both types of mutations were critical, as reversion of either CDR or FWR mutations to the wild-type amino acids resulted respectively in a loss of affinity of CK138 and CK157 toward ELR$^+$ CXC chemokines (Fig. 3d). Moreover,

**Fig. 3** Frequency and distribution of mutations in crossreactive antibodies. **a** Box-and-whiskers graph comparing the total number (left y-axis) and frequencies (right y-axis) of mutated residues detected in CK1-, CK2-, and CK4-derived clones before (light gray) and after (dark gray) the first (I) and second (II) process of selection, respectively. "Inp" indicates sequenced clones picked from the random yeast-display antibody library before selection (input). "Out" indicates sequenced clones after selection (output). "n" indicates samples size. The middle line within each box represents the median, and the lower and upper boundaries of the box indicate the 25th (Q1) and 75th (Q3) percentiles. Whiskers represent the 1.5× interquartile range (IQR = Q3–Q1) extending beyond box. Statistical comparisons were made between each group using one-way analysis of variance (ANOVA), followed by Tukey's test to calculate P-values: *P < 0.05, **P < 0.01, ***P < 0.001; ****P < 0.0001. ns: non-significant. Homology model and frequencies of enriched mutations of engineered **b** CK138 and **c** CK157 antibodies. Left, the $V_L$ and $V_H$ backbones are represented as ribbons (light gray). Mutations acquired during the selection process are depicted as spheres at the Cα positions. Mutated amino acids belonging to CDR loops of CK138 and CK157 are colored in dark blue and dark red, respectively. Diversified amino acids belonging to FWR regions of CK138 and CK157 are colored in light blue and light red, respectively. Right, columns graph reporting the mutation frequency in CDR (dark gray) and FWR (light gray) regions. Only amino-acid mutations of CK1 and CK2 lineages that showed at least 20% frequency and were enriched through two iterative processes of selection are reported. Wild type and mutated amino acids are listed at the top and bottom, respectively. **d** Fluorescence binding signal of CK1- (left) and CK2- (right) derived clones bearing highly frequent mutations within the CDR-H3 and FWRs, respectively, that were reverted to the wild-type amino acids. ELR$^+$ CXC chemokines and the corresponding binding/display values (y-axis) are indicated as differently colored filled circles and represent the means of at least three independent experiments

these mutations were found throughout different clones and cycles of engineering, suggesting strong selection pressure for these residues in conferring high crossreactivity and affinity (Supplementary Figs. 3a and 4a).

**Engineered antibodies bind a large array of chemokines**. To assess the extent of crossreactivity of the engineered antibodies, we characterized their binding affinity towards 20 human and murine CXC chemokines. The chemokine panel included 12 human and mouse ELR$^+$ CXC chemokines (which share 32–90% sequence identity) and 8 human and mouse ELR$^-$ CXC chemokines (which share 18–70% sequence identity). The ELR$^+$ CXC chemokines share 20–51% sequence identity with the ELR$^-$ CXC chemokines (Supplementary Fig. 7a). In order to accurately determine the $K_D$ values of these antibodies for the different chemokines, we utilized two complementary configurations of chemokines and antibodies in the context of yeast surface display (Supplementary Fig. 7b). Specifically, we performed titrations using (i) soluble CXC chemokines with yeast-displayed antibodies and (ii) soluble antibodies with yeast-displayed CXC chemokines (Supplementary Fig. 7c, d). Exploring both orientations was necessary as some CXC chemokines are known to form oligomers when present in high concentrations in solution, leading to undesired multivalent binding phenomena. We produced the CXC chemokines as fusions to the N-terminus of mouse SA (Supplementary Fig. 1e–h) and the engineered CK129, CK138, and CK157 single-chain variable antibody fragments as fusion to the C-terminus of SA, which are referred to as SA129, SA138, and SA157* (Supplementary Fig. 8 and Supplementary Table 4). SA157* is denoted with an asterisk as instead of being produced as a single chain with a linker, it was produced as separate $V_L$ and $V_H$ domains and mixed in equimolar amounts. In both orientations, we observed similar crossreactivity of our engineered antibodies towards CXC chemokines that were not included in the selection cycles (Fig. 4a). Yeast-displayed CK129, CK138, and CK157 bind 7, 12, and 16 soluble CXC chemokines, respectively (Fig. 4a and Supplementary Fig. 7f). Similarly, the soluble SA129, SA138, and SA157* bind 4, 11, and 14 yeast-displayed CXC chemokines, respectively (Fig. 4a, b). With a few exceptions, the $K_D$ values determined by using soluble SA129, SA138, and SA157* antibody fusions with yeast-displayed CXC chemokines were on average 2- to 5-fold higher than those measured in the opposite arrangement (Supplementary Fig. 7g and Supplementary Table 5). This discrepancy in measured $K_D$ values and extent of crossreactivity between the two assay orientations was not surprising and may reflect oligomeric CXC chemokines interacting with multiple yeast-displayed antibodies, thus confounding avidity effects with higher affinity. This phenomenon appears to be pronounced for ELR$^-$ CXC chemokines, such as hCXCL10 and hCXCL4, that are known to form highly avid oligomers in solution[31].

We also observed that SA129, which only recognizes four chemokines that share significant sequence identity, displays relatively high affinity for those targets. In contrast, highly crossreactive SA138 and SA157* had overall lower binding affinities toward a larger array of targets, suggesting an inverse correlation between affinity and extent of crossreactivity (Fig. 4c and Supplementary Fig. 7h).

To assess that the crossreactivity was not merely due to non-specific polyreactivity, we characterized the binding of our engineered antibodies towards five structural related CC chemokines and eleven structurally unrelated proteins, including the chemotactic factor C5a (Supplementary Fig. 7e). No or weak binding was detected in the case of CK129 and CK138, confirming their specificity toward members of the CXC

chemokine family. Contrariwise, CK157 exhibits crossreactivity toward C5a and some (CCL20, CCL22, CCL28) but not all CXC and CC chemokines, suggesting a more promiscuous mode of binding.

**Crossreactive antibodies recognize functional epitopes**. We next performed fine epitope mapping using alanine-scanning mutagenesis to identify the residues that are directly involved in the interactions[32]. hCXCL1 was chosen as our model chemokine over other ELR$^+$ CXC chemokines because it is recognized by all the engineered crossreactive antibodies and is well-characterized biochemically[33,34]. We first combined three-dimensional structural analysis and literature data to identify hCXCL1 amino acids suitable for mutagenesis. Structurally buried hydrophobic amino acids and proline and cysteine residues were left unaltered, as they are crucial for overall chemokine folding and stability (Supplementary Fig. 9a). Fifty-four predicted solvent-exposed hCXCL1 residues were selected, individually mutated to alanine, expressed on the surface of yeast (Supplementary Fig. 9b), and screened for decreased binding affinity to the soluble SA129, SA138, and SA157*. Five mutants that exhibited an intense loss of binding upon incubation with all three antibodies were excluded, as this phenomenon was likely due to protein misfolding and destabilization of the displayed variants (Supplementary Fig. 9c). We then assessed the binding of the remaining 49 hCXCL1 mutants to soluble SA129, SA138, and SA157*. Solvent-exposed mutations that eliminated or significantly reduced binding affinity were identified, which allowed us to pinpoint residues that were likely critical for the interactions (Fig. 5a and Supplementary Table 6). We further identified the epitopes of two commercially available neutralizing antibodies: the highly specific Ab275 (binds only hCXCL1) and the crossreactive Ab276 (binds hCXCL1, hCXCL2, and hCXCL3). Next, we compared their epitope maps with the maps assigned to our engineered antibodies (Fig. 5a and Supplementary Table 6). Similarly to Ab275 and Ab276, SA129 and SA138 bind motifs along the functional N- and 40s-loops that are known to be crucial for the binding of hCXCL1 to its cognate receptor, CXCR2. In contrast, SA157* recognizes a distinctive epitope and engages binding with hCXCL1 residues that are important for its interaction with glycosaminoglycans (GAGs; Fig. 5b). These epitope maps are also consistent with results from a competitive assay (Supplementary Fig. 9d). Notably, the residues recognized by the highly crossreactive SA138 and SA157* are conserved among many different CXC chemokines, thus explaining these antibodies' broad promiscuity (Fig. 5c).

We also observed that the total number of residues involved in binding and the frequency of interaction strengths are different across the characterized antibodies. The relatively more specific Ab275, Ab276, and SA129 engage binding with a larger number of hCXCL1 residues than the more crossreactive SA138 and SA157*, and strong interactions are more frequent. In contrast, the binding specificity of SA138 and SA157* appear to be achieved mostly through weak interactions toward a few conserved residues (Supplementary Fig. 9e, f). These observations are consistent with affinity measurements, which show that the broadly crossreactive SA138 and SA157* display lower affinity toward their targets compared to the more specific SA129. Taken together, epitope mapping and competition assays revealed that our engineered crossreactive molecules recognize distinct conserved epitopes on CXC chemokines. Data also suggest that crossreactivity towards a large array of chemokines is achieved through weak recognition of a few highly conserved residues.

**Crossreactive antibodies inhibit chemokine-binding in vitro**. As a measure of potential therapeutic efficacy, we then tested the

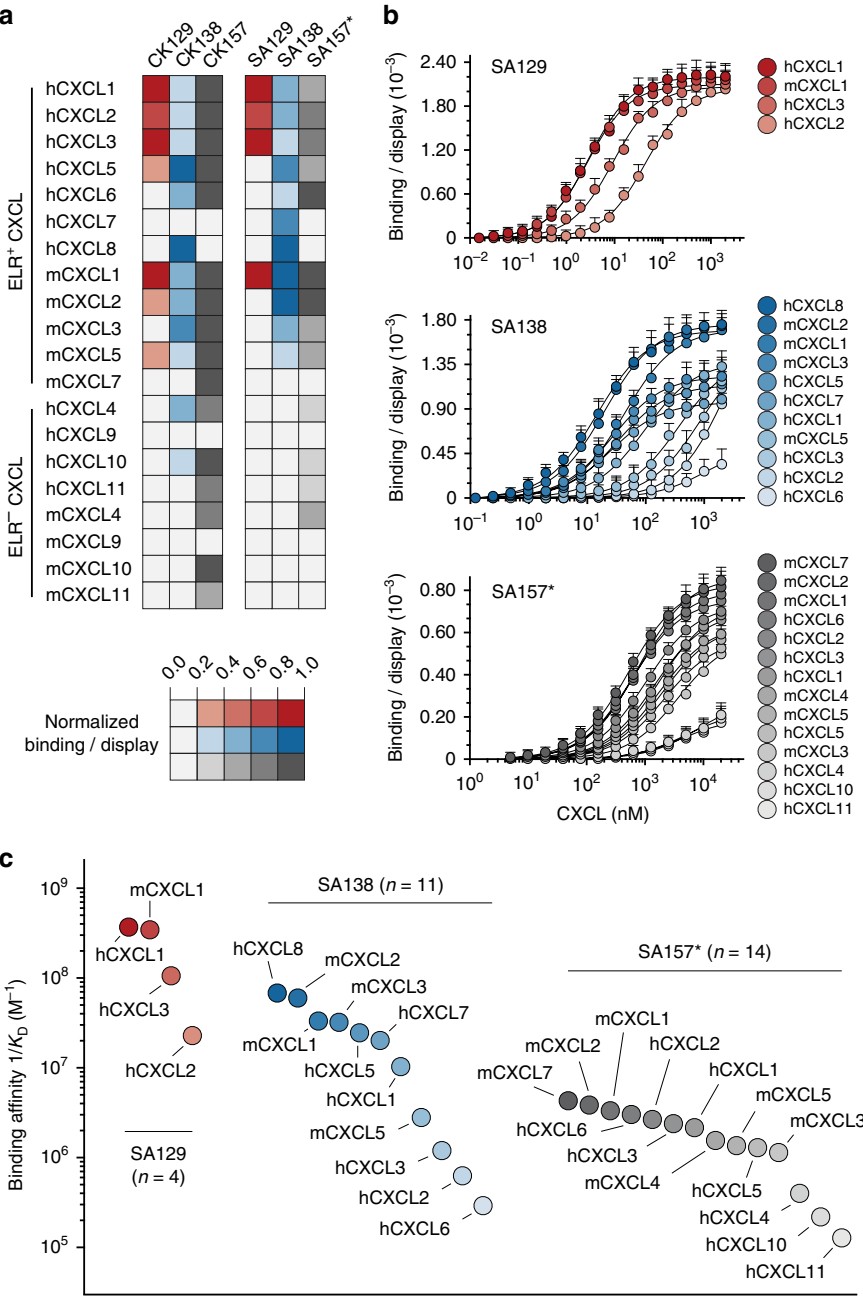

**Fig. 4** Crossreactivity of engineered antibodies toward multiple CXC chemokines. **a** Heat map indicating the binding intensity of the engineered antibodies against 20 diverse human and murine CXC chemokines. Binding was assessed by flow cytometry using two cell-display arrangements: soluble CXC chemokine against yeast-displayed CK129 (red), CK138 (blue), and CK157 (gray) antibodies (on the left) and soluble serum albumin–antibody fusions SA129 (red), SA138 (blue), and SA157* (gray) against yeast-displayed CXC chemokines (on the right). Normalized binding/display signal intensities range from light to dark colors indicating low (0.0–0.2) and high (0.8–1.0) titers, respectively. **b** Binding isotherms of yeast-displayed CXC chemokines to soluble serum albumin–antibody fusions SA129, SA138, and SA157*. Equilibrium binding affinity ($K_D$) values were determined only for chemokines exhibiting signals at high concentrations of soluble antibody fusions. CXC chemokines are gradient colored ranging from dark (high affinity) to light (low affinity) red (SA129), blue (SA138), and gray (SA157*). Data are presented as mean (dots) ± s.e.m. (bars). **c** Plot showing binding affinities of yeast-displayed CXC chemokines to SA129 (red), SA138 (blue), and SA157* (gray) antibody fusions. The indicated values are displayed as differently colored filled circles and represent the means of at least three independent experiments presented as inverse equilibrium binding constants ($1/K_D$; M$^{-1}$)

ability of crossreactive SA129, SA138, and SA157* antibody fusions to inhibit binding of ELR$^+$ CXC chemokines to their cognate CXCR1 and CXCR2 receptors. To this end, we used HEK293 cell lines expressing human CXCR1 and CXCR2 (Supplementary Fig. 10a, b). We then incubated the cells with various concentrations of hCXCL1 and hCXCL8 ligands to determine the half-maximal effective concentrations (EC$_{50}$) of the

interactions (Supplementary Fig. 10c and Supplementary Table 7). Next, we examined the ability of SA129, SA138, and SA157* to antagonize the interactions between hCXCL1 and hCXCL8 ligands and their cognate receptors. Our engineered antibodies inhibited the ability of human hCXCL1 and hCXCL8 chemokines to bind CXCR1 and CXCR2 receptors in a dose-dependent manner to various extents

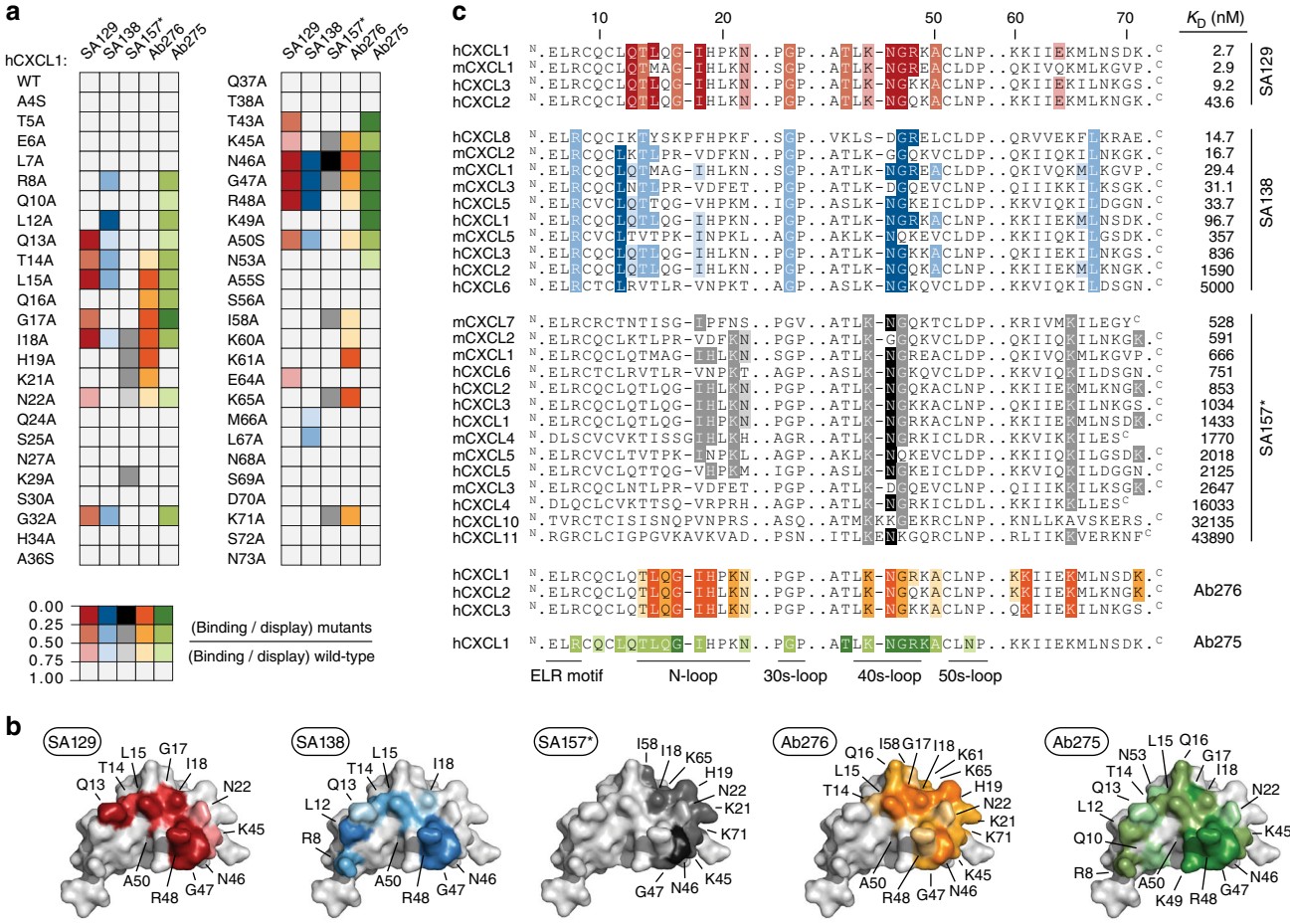

**Fig. 5** Epitope mapping of crossreactive antibodies. **a** Binding of SA129 (red), SA138 (blue), SA157 (gray), Ab275 (green), and Ab276 (orange) to a defined panel of hCXCL1 alanine mutants was assessed by flow cytometry. Obtained median values were normalized to the display median fluorescence intensities of each single yeast surface displayed mutant (binding/display). Normalized values represent the means of at least three independent experiments. Mutations that do not significantly affect binding (0.75–1.0) are shown in white, while mutations that weakly (0.5–0.75), moderately (0.25–0.5), or strongly (0.0–0.25) disrupt binding are shown respectively in light, intermediate, and dark colors. **b** The identified contact residues of hCXCL1 (PDB ID: 1MGS) to each antibody as defined by epitope mapping are shown in red (SA129), blue (SA138), gray (SA157*), green (Ab275), and orange (Ab276). The color intensity correlates with the strength of the interaction, with weak and strong interactions shown as light and dark colors, respectively. **c** Sequence alignment of various CXC chemokine proteins. Positions of conserved solvent-exposed residues that appear to be involved in the interaction with SA129 (red), SA138 (blue), SA157* (gray), Ab275 (green), and Ab276 (orange) based on residues identified using hCXCL1 alanine mutants are shown. Amino-acid sequences have been listed based on binding affinity ($K_D$), with the tightest CXC chemokine protein at the top and the weakest at the bottom. Upper case N and C letters indicate the N- and C-terminus of the amino-acid sequence, respectively. Regions including residues that are not involved in binding are not reported for space reasons. The regions denoting the ELR-motif, N-loop, 30s-loop, 40s-loop, and 50s-loop that are known to be crucial for the binding of ELR$^+$ CXC chemokines to the cognate CXCR2 receptor are indicated at the bottom. Residues have been highlighted according to the strength of interaction determined using soluble antibodies against hCXCL1 alanine mutants, as shown in panel **a**

(Supplementary Fig. 10d and Supplementary Table 8). Remarkably, the determined inhibitory constants ($K_i$) correlated well with the previously reported $K_D$ values (Supplementary Fig. 10e). These results showed that crossreactive SA129, SA138, and SA157* can interfere with the binding of ELR$^+$ CXC chemokines to both human CXCR1 and CXCR2 in vitro.

We further assessed the ability of the SA129, SA138, and SA157* to antagonize the activation of ELR$^+$ CXC chemokine receptors. For this purpose, we utilized an intracellular calcium mobilization assay in the presence of human- and mouse-derived neutrophils activated respectively with human (hCXCL1, hCXCL5, and hCXCL8) and murine (mCXCL1 and mCXCL2) ELR$^+$ CXC chemokines. To this end, we first determined the EC$_{50}$ of these chemokines on the neutrophils (Supplementary Fig. 10f and Supplementary Table 9). Then, we monitored changes in intracellular calcium levels upon pre-incubation of ELR$^+$ CXC chemokines with varying concentrations of SA129,

SA138, and SA157* as antagonists. Commercial neutralizing monoclonal antibodies were also used as a positive control. The assays revealed that our engineered antibodies exhibited encouraging inhibitory activity by preventing binding of the human and murine ligands to their receptors in a dose-dependent manner (Fig. 6a, b and Supplementary Table 10). Again, the calculated $K_i$ values correlated well with the previously determined $K_D$ affinities (Fig. 6c). Taken together, these data indicate that engineered crossreactive molecules are able to inhibit ELR$^+$ CXC chemokine signaling in vitro and ex vivo, and have the promise to suppress CXCR1 and CXCR2 activation in vivo.

**SA–antibody fusion reverses inflammation in vivo.** Given the promising results from our inhibitory assays, we then tested the effect of our engineered antibody fusions in the K/BxN serum transfer model of autoantibody-induced arthritis[35], which

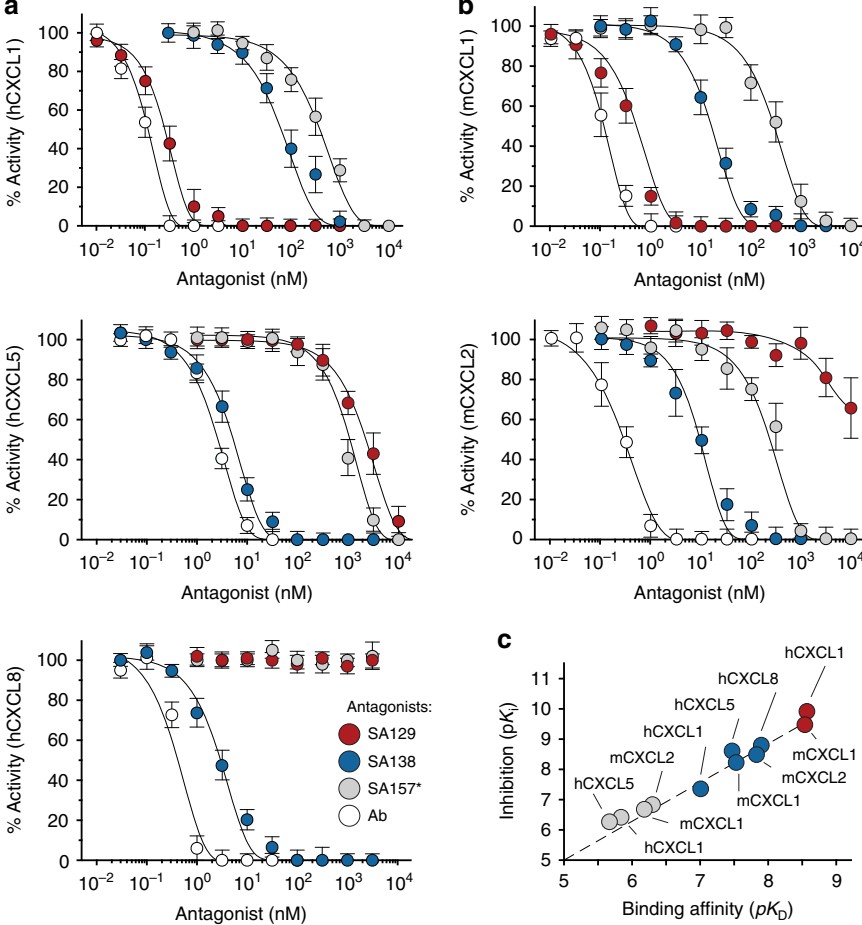

**Fig. 6** Crossreactive antibodies inhibit ELR$^+$ CXC chemokine signaling in vitro. Residual activity of **a** human hCXCL1, hCXCL5, and hCXCL8 and **b** mouse mCXCL1 and mCXCL2 chemokines incubated with varying concentrations of SA129 (red), SA138 (blue), and SA157* (gray) fusions, and commercial neutralizing antibodies (Ab, white). The indicated values are the means of three independent experiments. **c** Plot displaying p$K_i$ versus the calculated p$K_D$ of SA129 (red), SA138 (blue), and SA157* (gray) fusions. Data are presented as mean (dots) ± s.e.m. (bars)

displays clinical and histopathological similarities to human RA[36]. The levels of ELR$^+$ CXC chemokines are elevated in the joints of this arthritic mice and neutrophils, which have upregulated CXCR2 in the joint, are the main effector cells, making the K/BxN serum transfer-induced arthritis mice a valid model to test the therapeutic efficacy of our engineered binders[13,14,37,38]. To antagonize circulating ELR$^+$ CXC chemokines in vivo, we generated SA–antibody fusions (Supplementary Fig. 8). Fusion of our engineered antibody fragments to SA improved solubility and stability. In addition, SA is non-immunogenic, and the fusion molecules could be produced in high yields conducive to high drug doses[39]. Furthermore, similarly to full-length antibody, SA exploits the FcRn receptor to achieve a prolonged circulatory half-life. Its plasma persistence is nonetheless still shorter than that of full-length monoclonal antibodies, thus circumventing the "buffering" effects that are experimentally observed with the use of antibodies targeting small antigens[40–43]. Unlike an antibody, SA does not bind the FcγR receptors expressed on immune cells, thus precluding additional immune activation and inflammation mediated by antibody-dependent cell-mediated cytotoxicity (ADCC)[44]. Finally, SA has been shown to accumulate at high levels in inflamed joints, making it a promising drug carrier for RA[45]. For this study, we used the SA129 and SA138 fusions described earlier. We excluded SA157* because of its limited solubility and crossreactivity towards non ELR$^+$ CXC chemokines (Supplementary Figs. 7e and 8d). As a negative control, we

used an irrelevant SA fusion (SA$^{CTR}$) encoding SA fused to an antibody fragment that targets the human carcinoembryonic antigen (CEA), a protein that does not exist in mice[46]. To ensure complete inhibition of all ELR$^+$ CXC chemokines present in circulation, we used fairly high doses of our engineered antibody fusions (50 mg kg$^{-1}$). When injected into mice, SA129, SA138, and SA$^{CTR}$ displayed plasma half-lives between 42 and 47 h, considerably longer than that of small synthetic compounds or antibody fragments, but still shorter than that of full-length monoclonal antibodies (Supplementary Fig. 11a and Supplementary Table 11). Despite the high doses of SA129, SA138, and SA$^{CTR}$ used, the molecules were well tolerated. Treated mice gained weight and exhibited good body condition (Supplementary Fig. 11b). We initially assessed the ability of crossreactive antibody fusions to prevent the manifestation of inflammatory arthritis in the K/BxN serum transfer model. Mice were treated on the same day that the arthritogenic serum was injected (Supplementary Fig. 11c), and disease progression was evaluated by both blinded clinical scores and measurements of ankle thickness. Mice treated with the more crossreactive SA138, which binds all four murine ELR$^+$ CXC chemokines (mCXCL1, mCXCL2, mCXCL3, and mCXCL5), were protected from developing arthritis, with an approximately 80% reduction of clinical score compared with negative controls at the peak of the disease (day 8 after arthritogenic K/BxN serum transfer and disease initiation). In contrast, the more specific SA129 that recognizes

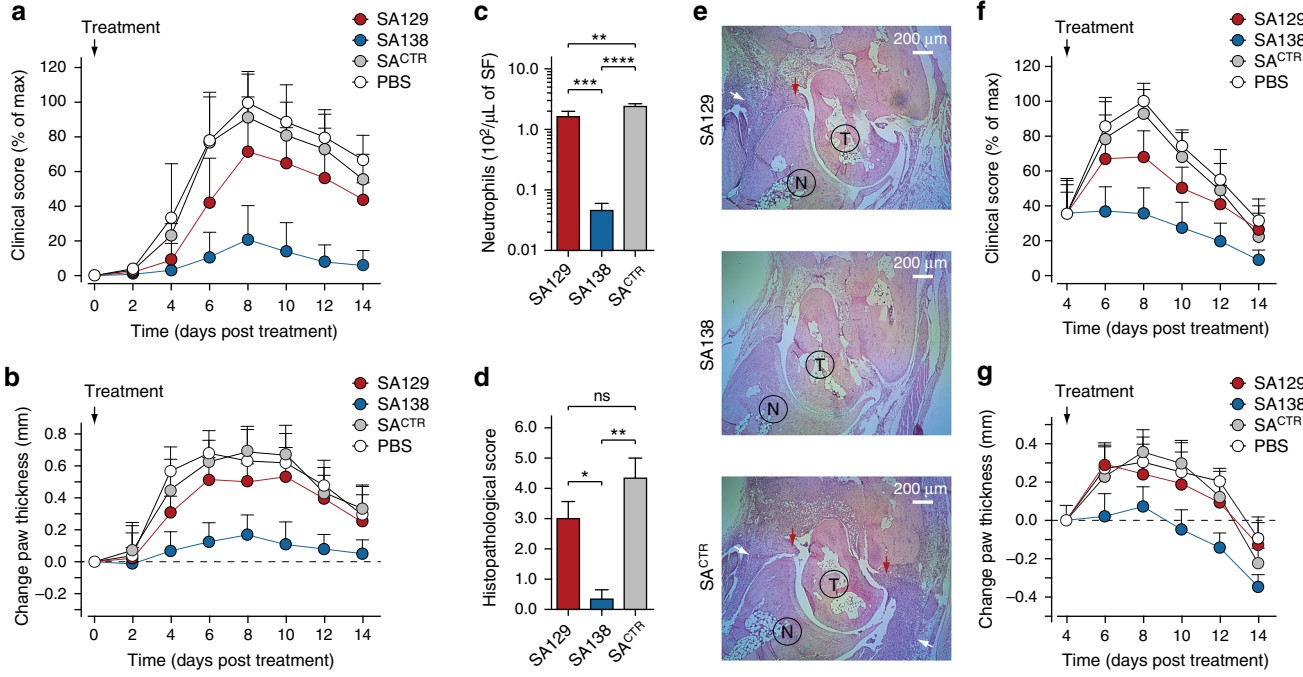

**Fig. 7** Crossreactive serum albumin–antibody fusion reverses inflammation in vivo. **a** Clinical score (% of max) and **b** change in ankle thickness (mm) of mice treated with serum albumin–antibody fusion proteins on day 0 (preventative regimen). Arthritogenic serum was injected into C57BL/6J on days 0 and 2. Mice were also treated daily with SA129, SA138, and SA$^{CTR}$ fusions (1 mg per mouse in PBS i.p.) beginning on day 0. Paw thickness of ten mice per group ($n = 10$), pooled from two independent experiments, were measured every 2 days for a total of 14 days. Arrows indicate first day of treatment. Data are presented as mean (dots) ± s.e.m. (bars). **c** Columns graph reporting the number of infiltrating synovial fluid neutrophils (Ly6G$^+$ cells) from the ankles of serum-transferred arthritic mice measured at day 8 by flow cytometry ($n = 3$ per condition). Statistical comparisons were made between each group using one-way analysis of variance (ANOVA), followed by Tukey's test to calculate $P$-values: *$P < 0.05$, **$P < 0.01$, ***$P < 0.001$; ****$P < 0.0001$. ns: non-significant. **d** Columns graph reporting the histopathological scoring and **e** representative H&E staining of ankle tissue sections of mice treated with SA129 (top), SA138 (middle), and control SA$^{CTR}$ (bottom) on day 8. Scale bar represents 200 μm. White arrows indicate joint-infiltrating inflammatory cells, and red arrows indicate pannus formation. T taulus, N navicular. **f** Clinical score (% of max) and **g** change in ankle thickness (mm) of K/BxN serum-induced arthritic mice treated beginning on day 4 with serum albumin–antibody fusion proteins (therapeutic regimen). Arthritogenic serum was injected into C57BL/6J on days 0 and 2, and mice were then treated daily i.p. with SA129, SA138, and SA$^{CTR}$ fusions (1 mg per mouse in PBS i.p.) beginning on day 4 after inflammation had developed. Paw thickness of ten mice per group ($n = 10$), pooled from two independent experiments, was measured every 2 days for a total of 14 days. Arrows indicate the day treatment began. Data are presented as mean (dots) ± s.e.m. (bars)

just one murine ELR$^+$ CXC chemokine (mCXCL1) only moderately reduced joint inflammation, with an approximately 30% reduction of clinical score at day 8. Mice treated with SA$^{CTR}$ showed the typical clinical signs of untreated mice that received arthritogenic serum and developed inflammatory arthritis with pronounced joint swelling. There were no differences between mice treated with SA$^{CTR}$ or with vehicle (PBS) only (Fig. 7a, b).

We next quantified the number of synovial fluid neutrophils isolated from the arthritic joints of mice treated with SA129, SA138, and SA$^{CTR}$ antibody fusions. Synovial fluids were harvested at the peak of the disease (day 8 after disease initiation). We observed that mice treated with the broadly crossreactive SA138 had, respectively, 50- and 70-fold lower levels of synovial fluid neutrophils than mice treated with the more specific SA129 or the irrelevant SA$^{CTR}$ (Fig. 7c). These data are consistent with previous clinical score measurements and resembled those observed for CXCR2-deficient mice ($Cxcr2^{-/-}$) injected with arthritogenic serum[13,14]. We also performed histological analysis and scoring of inflamed ankle sections. Inflammatory cell infiltration and pannus formation were absent or minimally present in mice treated with the broadly crossreactive SA138 (Fig. 7d, e). Consistent with previous clinical findings, the joints of mice treated with arthritogenic serum and control SA$^{CTR}$ displayed abundant inflammatory cell infiltration and pannus formation. These pathological changes were present, though less pronounced, in mice treated with the more specific SA129. In addition, the number of von Willebrand factor (vWF)-positive endothelial cells (ECs) in the joint tissue of mice treated with SA138 was lower than the number of vWF-positive ECs in the joint tissue of mice treated with SA129 or SA$^{CTR}$, suggesting that SA138 also prevented angiogenesis (Supplementary Fig. 12).

Finally, we tested the therapeutic efficacy of crossreactive antibody fusions in mice with established arthritis. Arthritic mice were treated 4 days after arthritogenic serum transfer, when joint inflammation had developed (Supplementary Fig. 11d). The highly crossreactive SA138 rapidly reversed inflammation and improved disease outcome with nearly 60% reduction of clinical score and 0.3 mm of ankle thickness over control at the peak of the disease (day 8 after disease initiation). The specific SA129-treated mice exhibited only a modest reduction of both clinical scores (~25%) and ankle thickness (0.1 mm) at day 8. The SA$^{CTR}$ and vehicle-treated mice showed no difference in the rate of disease development (Fig. 7f, g). Taken together, these data show that the broadly crossreactive SA138 molecule is capable of blocking neutrophil infiltration in the synovial tissues, thus preventing and even reversing inflammatory arthritis.

## Discussion

Chronic inflammatory diseases usually involve multiple ligands and receptors acting in concert. As a result, therapies targeting a

single pathological factor are often insufficient to achieve desired clinical outcomes, as has proven to be the case with anti-chemokine therapies. To overcome this limitation, we engineered highly crossreactive proteins capable of simultaneously blocking multiple human and murine ELR$^+$ CXC chemokines.

Highly crossreactive antibodies are challenging to obtain using traditional methodologies involving animal immunization and hybridoma development. The immune system tends to remove self-reactive antibodies, making it difficult to generate in vivo antibodies against sequence- and structurally related antigens derived from different species. In contrast, in vitro antibody libraries associated with display technologies are unaffected by immune tolerance[47]. Here we applied a multiple-pressure iterative combinatorial approach for the isolation of chemokine-blocking antibodies with broad crossreactivity. Even though none of the engineered antibodies could recognize with high affinity all members of ELR$^+$ CXC chemokine family, this still represents, to the best of our knowledge, the first systematic study reporting the selection strategy for the in vitro directed evolution of binders with such extensive promiscuity towards a panel of structurally related, yet sequence-diverse, protein targets. Nevertheless, our in vitro engineering approach is not without limitations. Broadly crossreactive antibodies were complex to develop, often had weak binding affinities and greater number of mutations compared with the more specific antibodies, which could lead to potential instability and immunogenicity.

The biochemical characterization of crossreactive binders isolated in this study allowed us to uncover some interesting features. We observed that the binding mode of in vitro evolved crossreactive CK138 and CK157 antibodies appear to resemble those found in naturally existing crossreactive ELR$^+$ CXC chemokine-binding proteins. First, both the length and the amino-acid composition and distribution of the CDR-H3 loop of the crossreactive CK138 resembles that of the N-terminal extracellular binding loop of the promiscuous CXCR2 receptor[48]. Moreover, CK138 binds ELR$^+$ CXC chemokine residues that are known to be recognized by that same N-terminal extracellular loop, supporting the idea that both promiscuous molecules share similar binding mode (Supplementary Fig. 13a). Second, the crossreactive CK157 accumulated numerous surface-exposed FWR mutations at the dimerization interface between the $V_L$ and $V_H$ chains. These mutations resulted in the replacement of positive charges with negatively charged residues, allowing the development of a defined structural patch that resembles the one present in the naturally evolved highly crossreactive viral chemokine-binding proteins (vCKBPs)[49,50]. Epitope mapping data revealed the ability of the highly crossreactive CK157 to recognize the positively charged GAG-binding residues of ELR$^+$ CXC chemokines. This supports the hypothesis that the binding mode of CK157 could be similar to that of vCKBPs, where a large negatively charged area engages conserved GAG-binding regions on CC and CXC chemokine proteins, thus explaining their extensive promiscuity (Supplementary Fig. 13b).

In vitro evolved crossreactive antibodies also share some features present in other crossreactive proteins of the immune system[51]. Indeed, similar to naturally occurring broadly neutralizing antibodies, the in vitro evolved crossreactive antibodies accumulated a high number of mutations during the selection process, and in the case of CK157, mutations were preferentially located in FWRs[52]. Moreover, in vitro evolved crossreactive antibodies appear to achieve binding to large array of diverse proteins by engaging a small number of energetically favored hot-spots (typically structurally and chemically conserved) surrounded by weaker and more diverse peripheral interactions. This mechanism of interaction resembles that of αβ T-cell antigen receptors (TCR), which bind multiple diverse short peptides presented by

major histocompatibility complex (MHC) molecules[53,54]. Nevertheless, other molecular mechanisms could also contribute to high crossreactivity, such as (i) structural flexibility, (ii) molecular complementarity, (iii) entropic contributions, and (iv) the chemical composition of both binding (paratope) and recognition (epitope) sites (Supplementary Discussion)[55–59].

Using the engineered crossreactive molecules described in the present study, we proposed a SA–antibody fusion-based strategy to enable optimal pharmacokinetic profiles, thus overcoming buffering effect phenomena that have limited previous interventions. Importantly, when these fusions were tested in the murine K/BxN serum transfer model of inflammatory arthritis, broadly crossreactive SA138 demonstrated greater therapeutic efficacy than the more specific molecule SA129, confirming that blocking multiple neutrophil-active ELR$^+$ CXC chemokines is required to completely inhibit neutrophil infiltration into the joints and both prevent and resolve inflammatory arthritis in vivo. Future research efforts should be oriented toward testing their efficacy in other models of arthritis in order to assess their therapeutic potential for the treatment of RA. In addition to inflammatory disease, we envisage that usage of such a molecule could augment the efficacy of current cancer treatments. Synergistic therapeutic antitumor immunity effects of CXCR2 blockade combined with chemotherapy[60] or anti-PD1 immunotherapy[61,62] have recently been shown.

In conclusion, we report here the in vitro engineering of broadly crossreactive chemokine-blocking antibodies with promising inhibitory effects in vivo and provide a basis for the development of next-generation therapeutics. We developed this concept with crossreactive antibodies against ELR$^+$ CXC chemokine ligands, but these studies also have value as a proof-of-concept for a general approach that could be applied to other therapeutic interventions targeting multiple soluble factors. Although many challenges still remain, the ability to evolve promiscuous and species-crossreactive proteins using directed evolution techniques might provide not only superior therapeutic efficacy, but also better assessment of treatment toxicity as well as simpler and lower cost clinical trials, by facilitating the transition from pre-clinical models to clinical studies in humans.

## Methods

**Cloning of chemokine fusion proteins.** CXC chemokines were cloned as C-terminal fusion of the immunoglobulin fragment crystallizable (Fc) domain ($^N$Fc-CXCL$^C$) and as N-terminal fusion of the murine SA protein ($^N$CXCL-SA$^C$). All mammalian expression vectors are based on gWiz (Genlantis) containing an optimized human cytomegalovirus promoter and a kanamycin antibiotic resistance gene. Constructs for expression of $^N$Fc-CXCL$^C$ fusion proteins were generated by using a modified *Pfu* DNA polymerase-mediated site-directed mutagenesis protocol[63]. *Pfu*Ultra II Fusion HS DNA polymerase was obtained from Agilent Technologies, *Dpn*I enzyme from New England BioLabs, and the oligonucleotide primers from Integrated DNA Technologies. The synthetic DNA coding for the active form of three highly diverse human (hCXCL1, hCXCL5 and hCXCL8) and murine (mCXCL1, mCXCL2 and mCXCL5) ELR$^+$ CXC chemokines were obtained from GeneArt Gene Synthesis (Thermo Fisher Scientific). Genes were codon-optimized for expression in mammalian cells. A sequence encoding for Gly–Gly dipeptide spacer (G$_2$, $^N$GG$^C$) followed by a 15 amino-acid peptide sequence (AviTag) containing a defined lysine for site-specific biotinylation ($^N$GLNDI-FEAQKIEWHE$^C$) were inserted at the C-terminus of the ELR$^+$ CXC chemokine to obtain $^N$CXCL-G$_2$-AviTag$^C$ synthetic genes. The AviTag sequence for enzymatic biotinylation was placed at the well tolerated C-terminus of the ELR$^+$ CXC chemokines to (i) preserve unaltered the functional N-terminus region, (ii) avoid loss of epitope recognition, and (iii) prevent additional structural heterogeneity that could be triggered by performing a chemistry-based amine-reactive succinimidyl esters based biotinylation. The de novo synthesized $^N$CXCL-G$_2$-AviTag$^C$ synthetic sequences were subsequently inserted into a previously modified gWiz expression vector containing a DNA sequence encoding for a secretory leader peptide sequence ($^N$MRVPAQLLGLLLLWLPGARC$^C$), an Fc domain derived from a murine IgG2 heavy-chain constant regions $C_H$2 and $C_H$3, followed by a sequence encoding a hexa-histidine tag (His$_6$; $^N$HHHHHH$^C$), an eight amino-acid flexible linker ($^N$SSGVDLGT$^C$) and a Tobacco Etch Virus proteolytic cleavage site (TEV; $^N$ENLYFQ|A/V$^C$) to obtain the final $^N$Fc-His$_6$-linker-TEV-CXCL-G$_2$-AviTag$^C$

fusion proteins. The His$_6$-tag was inserted between the Fc domain and the TEV cleavage site for further purification steps. Importantly, the TEV proteolytic cleavage site allows for a precisely processed N-terminus of the chemokines that is crucial for their activity. All constructs were verified by DNA sequencing (Macrogen) and termed $^N$Fc-CXCL$^C$ fusion proteins (see Supplementary Data 1 for information about protein accession numbers, oligonucleotide primers, DNA, and amino-acid sequences). Constructs for expression of $^N$CXCL-SA$^C$ fusion proteins were generated using DNA assembly methods such as Gibson Assembly (New England BioLabs) and In-Fusion Cloning (Clontech Laboratories, Takara Bio) technologies. PfuUltra II Fusion HS DNA Polymerase and Herculase II Fusion DNA Polymerase (Agilent Technologies) were used for the PCR amplification of the insert and the vector, respectively. DpnI enzyme was obtained from New England Biolabs and the oligonucleotide primers from Integrated DNA Technologies. The synthetic DNA coding for the active protein form of 12 human and murine ELR$^+$ CXC chemokines and 8 human and murine ELR$^-$ CXC chemokines were obtained from GeneArt Gene Synthesis. Genes were codon-optimized for expression in mammalian cells. The de novo synthesized $^N$CXCL$^C$ synthetic sequences were subsequently inserted into a previously modified gWiz expression vector containing a DNA sequence encoding for a secretory leader sequence ($^N$MRVPAQLLGLLLLWLPGARC$^C$), a ten amino-acid flexible linker ($^N$GGGGSGGGGS$^C$), sequence encoding for mouse SA followed by a sequence encoding for a five amino-acid flexible spacer ($^N$GGGGS$^C$) and a hexa-histidine tag (His$_6$; $^N$HHHHHH$^C$) to obtain $^N$CXCL-(G$_4$G)$_2$-SA-G$_4$S-His$_6^C$ fusion proteins. The process of the leader sequence during the secretory pathway allows for a precisely cleaved N-terminus that is crucial for the activity of the chemokines. Genes encoding $^N$CXCL-(G$_4$G)$_2$-SA-G$_4$S-His$_6^C$ fusion proteins were further sub-cloned into a new gWiz expression vector via SalI-HF (New England BioLabs) and MauBI (Thermo Fisher Scientific) restriction enzymes. All constructs were verified by DNA sequencing and termed $^N$CXCL-SA$^C$ fusion proteins (see Supplementary Data 2 for information about protein accession numbers, oligonucleotide primers, DNA, and amino-acid sequences).

**Cloning of SA–antibody fusions.** Selected crossreactive synthetic single light (V$_L$) and heavy (V$_H$) chain antibody variable fragments (scFv) were cloned and expressed in mammalian cells as C-terminal fusion of the murine SA protein ($^N$SA-scFv$^C$). Mammalian expression vectors are based on gWiz. Constructs for expression of $^N$SA-scFv$^C$ fusion proteins were generated using DNA assembly methods and enzymes described above. The DNA sequences encoding the scFv (V$_L$–V$_H$ orientation) CK129, CK138, and CK157 as well as separate V$_L$ and V$_H$ domains of CK157 were amplified in a PCR reaction by using the pCT-CON vector as a template and following inserted into a previously modified gWiz expression vector containing a DNA sequence encoding for a secretory leader peptide sequence ($^N$MDMRVPAQLLGLLLLWLPGARC$^C$) followed by a sequence encoding the mouse SA, a 15 amino-acid flexible linker ($^N$GGGGSGGGGSGGGGS$^C$). A sequence encoding for a five amino-acid flexible linker ($^N$GGGGS$^C$) followed by a hexa-histidine tag (His$_6$; $^N$HHHHHH$^C$) was inserted at the C-terminus of the gene encoding the scFv to obtain the final $^N$SA-(G$_4$S)$_3$-scFv-G$_4$S-His$_6^C$, $^N$SA-(G$_4$S)$_3$V$_L$-G$_4$S-His$_6^C$, and $^N$SA-(G$_4$S)$_3$-V$_H$-G$_4$S-His$_6^C$ fusion proteins. In a similar fashion, the control scFv (V$_H$–V$_L$ orientation) targeting the human CEA was fused at the C-terminus of mouse SA. The stability of each scFv was further improved by connecting the V$_L$ and V$_H$ domains via an intermolecular disulfide bond (ds). Two of the most favorable locations have been selected for the introduction of pairs of cysteine residues into each single scFv (ds1: V$_L$100 and V$_H$44; ds2: V$_L$43 and V$_H$105; Kabat numbering system) and their relative effects on expression, percent monomer formation, and retention of antigen binding compared. Cysteine residues were introduced into each scFv by site-directed mutagenesis using DNA assembly methods and standard oligonucleotide primers carrying single point mutations (Integrated DNA Technologies). Final genes encoding $^N$SA-(G$_4$S)$_3$-scFv-G$_4$S-His$_6^C$, $^N$SA-(G$_4$S)$_3$-scFv-ds1-G$_4$S-His$_6^C$, $^N$SA-(G$_4$S)$_3$-scFv-ds2-G$_4$S-His$_6^C$, $^N$SA-(G$_4$S)$_3$-V$_L$-G$_4$S-His$_6^C$, and $^N$SA-(G$_4$S)$_3$-V$_H$-G$_4$S-His$_6^C$ fusion proteins were further sub-cloned into a new gWiz expression vector via NotI-HF and XbaI (New England BioLabs) restriction enzymes. All constructs were verified by DNA sequencing (see Supplementary Data 3 for information about oligonucleotide primers, DNA, and amino-acid sequences). The SA–antibody fusion formats were used for all in vitro and in vivo studies.

**Production of fusion proteins.** SA ($^N$CXCL-SA$^C$ and $^N$SA-scFv$^C$) and Fc ($^N$Fc-CXCL$^C$) fusion proteins were expressed by transient transfection of suspension-adapted human embryonic kidney (HEK293) cells. Cell lines were not authenticated and were tested for mycoplasma contamination. Protein production was performed either in house using FreeStyle 293 Expression System (Thermo Fisher Scientific) or outsourced to the Protein Expression Core Facility (PECF) of the Life Science Faculty of the EPFL[64,65]. At the end of the 7-day phase production, cells were harvested by centrifugation at 15,000 × g for 30 min at 4 °C on an Avanti JXN-26 Centrifuge (Beckman Coulter). Any additional cell debris was removed from the medium by filtration through 0.22-μm PES membrane filters (Thermo Fisher Scientific).

**Purification of Fc fusion proteins.** Clarified cell culture medium was diluted with 1/10 volume 10× PBS pH 7.4. Recombinant Fc fusions $^N$Fc-CXCL$^C$ were captured on a rProtein A Sepharose Fast Flow resin (GE Healthcare), packed on a glass Econo-Column Chromatography column (Bio-Rad), that was previously equilibrated with 10 column volumes (CVs) of 1× PBS pH 7.4. The filtered culture media was passed through the resin at a flow rate of approximately 2.5 mL min$^{-1}$ at room temperature. The resin was then washed with 10 CVs of 1× PBS pH 7.4 and the recombinant Fc fusions eluted in a single peak by applying 10 CVs of elution Buffer E (50 mM glycine-HCl, pH 2.7). Two CVs of neutralizing Buffer N (1 M Tris-HCl pH 8.5) were then immediately added to the eluted Fc fusion proteins to prevent protein denaturation. Eluted Fc fusions were then diluted twice with 1× PBS pH 7.4 and concentrated by using 10000 NMWL Amicon Ultra-15 ultrafiltration devices (Millipore) at 4000 × g and 4 °C on a Allegra X-14R centrifuge (Beckman Coulter). The concentrated Fc fusion proteins were further subjected to size-exclusion chromatography (SEC) by using a Hiprep 26/10 desalting column (GE Healthcare) connected to an AKTApurifier system (GE Healthcare) equilibrated with Buffer T (50 mM Tris-HCl, 100 mM NaCl, 0.5 mM EDTA, pH 8.0). Purified Fc fusion proteins $^N$Fc-CXCL$^C$ in Buffer T were further concentrated to 2 mg mL$^{-1}$ by using 10000 NMWL Amicon Ultra-15 ultrafiltration devices at 4000 × g and 4 °C on a Allegra X-14R centrifuge and cleaved by using recombinant TEV protease (0.5 mg mL$^{-1}$). Fc fusion:TEV at a molar ratio of 100:1 were incubated at 4 °C for up to 24 h in Buffer T supplemented with a 10:1 ratio of reduced (GSH) to oxidized (GSSG) L-glutathione (50 mM Tris-HCl, 100 mM NaCl, 0.5 mM EDTA, 3 mM GSH, 0.3 mM GSSG, pH 8.0) and complete protease inhibitor cocktail (Roche). The further separation of matured cleaved CXC chemokines from the (i) Fc domain, (ii) uncleaved Fc-CXCL fusion, and (iii) recombinant TEV-His$_6$ protease was performed by loading the cleavage mixture on a Ni Sepharose excel affinity resin (GE Healthcare), packed on a glass Econo-Column Chromatography column, that was previously equilibrated with 10 CVs of Buffer X (50 mM sodium phosphate, 500 M NaCl, pH 8.0). The mixture was passed through the resin at a flow rate of approximately 1 mL min$^{-1}$ at room temperature and the flow-through containing cleaved $^N$CXCL-G$_2$-AviTag$^C$ proteins collected. The purified $^N$CXCL-G$_2$-AviTag$^C$ proteins were further concentrated by using a 3000 NMWL Amicon Ultra-15 ultrafiltration devices (Millipore) at 4000 × g and 4 °C on a Allegra X-14R centrifuge and subjected to SEC by using a HiLoad 16/600 Superdex 75 prep-grade column (GE Healthcare) equilibrated with biotinylation Buffer R (50 mM Bicine, pH 8.3) on an AKTApurifier system. Purified $^N$CXCL-G$_2$-AviTag$^C$ proteins in Buffer R were then concentrated to approximately 100 μM by using 3000 NMWL Amicon Ultra-4 ultrafiltration devices (Millipore) at 4000 × g and 4 °C on a Allegra X-14R centrifuge. Molecular weights were confirmed by reducing sodium dodecyl sulfate–polyacrylamide gel electrophoresis (SDS-PAGE) using NuPAGE 4–12% Bis-Tris Gels (Thermo Fisher Scientific) in 2-(N-morpholino)ethanesulfonic acid (MES) buffer followed by SimplyBlue SafeStain (Thermo Fisher Scientific) and imaged on the Typhoon Trio imager (GE Healthcare). Purified $^N$CXCL-G$_2$-AviTag$^C$ proteins migrated as a single band in SDS-PAGE, with apparent molecular masses of about 8–10 kDa.

**Enzymatic biotinylation of chemokines.** Biotinylation of purified chemokines $^N$CXCL-G$_2$-AviTag$^C$ was performed by using BirA enzyme (Avidity). Enzymatic reaction included 50 nmol $^N$CXCL-G$_2$-AviTag$^C$ protein in Buffer R, 12 μg of recombinant BirA enzyme (3 mg mL$^{-1}$; Avidity), 50 μM d-biotin, 10 mM ATP pH 7.2, and 10 mM MgOAc for a total volume of 1 mL. To ensure complete biotinylation, the reaction was incubated at 4 °C for 48 h with gentle shacking and jumped started every 12 h by adding 50 μL of Biomix-A (500 mM Bicine, pH 8.3; Avidity) and 50 μL of Biomix-B (100 mM ATP, 100 mM MgOAc, 500 μM d-biotin; Avidity) to the reaction mix. These conditions were sufficient for complete quantitative reaction yielding one product with expected molecular mass (Δ mass = 226 Da). Biotinylated $^N$CXCL-G$_2$-AviTag$^C$ proteins were further purified by using either reversed-phase high-performance liquid chromatography (RP-HPLC) or SEC. RP-HPLC was performed on a Vydac C18 column (Grace & Co.) connected to a Waters HPLC system (Waters). A flow rate of 1 mL min$^{-1}$ and a linear gradient was applied with a mobile phase composed of eluant A (99.9% v/v H$_2$O and 0.1% v/v TFA) and eluant B (99.9% v/v ACN and 0.1% v/v TFA). This step efficiently removed unbound small molecules such as free biotin and ATP along with the BirA enzyme. Purified and biotinylated $^N$CXCL-G$_2$-AviTag$^C$ proteins were lyophilized, dissolved in 1 × PBS pH 7.4 to a final protein concentration of approximately 100 μM, flash frozen in liquid nitrogen, and stored at −80 °C. Alternatively, biotinylated $^N$CXCL-G$_2$-AviTag$^C$ proteins were purified by SEC using a Superdex 75 10/300 GL column (GE Healthcare) equilibrated with 1× PBS pH 7.4 and connected to an AKTApurifier system. The final purified and biotinylated proteins were further concentrated by using 3000 NMWL Amicon Ultra-0.5 centrifugal filter units (Millipore) at 14,000 × g and 4 °C on a Eppendorf 5702 R centrifuge (Eppendorf) to a final protein concentration of approximately 100 μM, flash frozen in liquid nitrogen, and stored at −80 °C. After purification, the yield of pure and biotinylated $^N$CXCL-G$_2$-AviTag$^C$ proteins ranged from 1 to 5 mg L$^{-1}$ of culture. Molecular weights were confirmed by SDS–PAGE as described above. Biotinylated $^N$CXCL-G$_2$-AviTag$^C$ proteins migrated as a single band in SDS-PAGE, with apparent molecular masses of about 8–10 kDa.

**Mass spectrometric analysis.** The molecular mass of each ELR$^+$ CXC chemokine before and after biotinylation was determined with electrospray ionization mass spectrometry (ESI-MS) performed on a quadrupole-time-of-flight mass spectrometer (Q-TOF) coupled to a C$_3$ or C$_8$ reversed-phase HPLC column for desalting of protein samples. Both LC-MS Agilent 6520 ESI-Q-TOF (Agilent Technologies) and Waters LCT ESI-Q-TOF (Waters) systems, operated in a positive ionization mode, were used. Data were acquired, processed, and analyzed using the Agilent MassHunter (Agilent Technologies) or the MassLynx (Waters) software package. Mass spectrometry confirmed the correct mass of the purified biotinylated chemokines and showed that no un-biotinylated protein remained in the final sample.

**Purification of SA fusion proteins.** Clarified cell culture medium was diluted with 1/10 volume Buffer A (500 mM sodium phosphate, 5 M NaCl, pH 8.0). Recombinant SA fusions were captured on an Ni Sepharose excel affinity resin, packed on a glass Econo-Column chromatography column, that was previously equilibrated with 10 CVs of Buffer B (50 mM sodium phosphate, 500 M NaCl, pH 8.0). The medium was passed through the resin at a flow rate of approximately 2.5 mL min$^{-1}$ at room temperature. The resin was then washed with 10 CVs of Buffer B and the recombinant SA fusions eluted in a single peak by applying 10 CVs of Buffer C (50 mM sodium phosphate, 500 M NaCl, 500 mM Imidazole, pH 8.0). Eluted SA fusions were following diluted twice with Buffer B and concentrated by using 10000 NMWL Amicon Ultra-15 ultrafiltration devices at $4000 \times g$ and 4 °C on a Allegra X-14R centrifuge. The concentrated SA fusion proteins were further purified by SEC using a HiLoad 16/600 Superdex 200 prep-grade column (GE Healthcare) equilibrated with 1× PBS pH 7.4 on an AKTApurifier system. Purified SA fusion proteins in 1× PBS pH 7.4 were following concentrated to 5 mg mL$^{-1}$ ($^{N}$CXCL-SA$^{C}$) and 2 mg mL$^{-1}$ ($^{N}$SA-scFv$^{C}$) final concentration by using 10000 NMWL Amicon Ultra-15 ultrafiltration devices at $4000 \times g$ and 4 °C on a Allegra X-14R centrifuge. Protein concentrations were determined by measuring absorbance at 280 nm using a NanoDrop 2000 spectrophotometer (Thermo Fisher Scientific). Molecular weights were confirmed by reducing SDS-PAGE using NuPAGE 4–12% Bis-Tris Gels in 3-(N-morpholino)propanesulfonic acid (MOPS) buffer followed by SimplyBlue SafeStain and imaged on the Typhoon Trio imager. All purified SA fusion proteins migrated as a single band in SDS-PAGE with an apparent molecular mass of approximately 75 kDa (for $^{N}$CXCL-SA$^{C}$), 80 kDa ($^{N}$SA-V$_{L}$$^{C}$ or $^{N}$SA-V$_{H}$$^{C}$), and 95 kDa ($^{N}$SA-scFv$^{C}$). The monodisperse state of concentrated SA fusion proteins was confirmed by SEC using a Superdex 200 10/300 GL column (GE Healthcare) connected to an AKTApurifier system and equilibrated with 1× PBS pH 7.4. Purified SA fusion proteins were eluted as a single peak at elution volumes ($V_e$) that corresponds to apparent molecular masses ranging between 150 kDa (dimer) and 300 kDa (tetramer) in the case of $^{N}$SA-CXCL$^{C}$ fusions while $^{N}$SA-scFv$^{C}$ fusions were eluted with $V_e$ that corresponds to apparent molecular masses of about 80 and 95 kDa (monomer). Size-exclusion chromatography columns and the FPLC system used for purification of $^{N}$SA-scFv$^{C}$ fusions for animal studies were pretreated with 1 M NaOH to remove endotoxins. Purified $^{N}$SA-scFv$^{C}$ fusions were further filtered sterile by passing them through a 0.2 μm syringe filters (Pall Life Sciences) and confirmed to contain minimal levels of endotoxin (<0.1 EU mL$^{-1}$) using the QCL-1000 Limulus Amebocyte Lysate (LAL) chromogenic test following the manufacturer's instructions (Lonza).

**Chemical biotinylation of proteins.** Reactive EZ-link sulfo-NHS-LC-biotin (Thermo Fisher Scientific) was dissolved in 1× PBS pH 7.4 to obtain a final concentration of 10 mM. Protein conjugates containing biotin were prepared by incubating SA fusion proteins (at concentrations of 2 mg mL$^{-1}$ in 1× PBS pH 7.4) with ten-fold molar excess of EZ-link sulfo-NHS-LC-biotin for 30 min at room temperature. Excess of unreacted or hydrolyzed biotinylation reagent was removed using SEC with Superdex 200 10/300 GL connected to an AKTApurifier system and equilibrated with buffer 1× PBS pH 7.4. Fractions corresponded to the expected protein pick were pulled and concentrated to a final concentration of 2 mg mL$^{-1}$ using 10000 NMWL Amicon Ultra-4 ultrafiltration devices (Millipore) at $4000 \times g$ and 4 °C on a Allegra X-14R centrifuge. Final protein concentrations were measured using a NanoDrop 2000 Spectrophotometer.

**Fluorescent labeling of proteins.** Reactive Alexa Fluor 647 succinimidyl ester (Thermo Fisher Scientific) was dissolved in anhydrous dimethylsulfoxide (DMSO; Sigma-Aldrich) to obtain a final concentration of 10 mg mL$^{-1}$. Protein conjugates containing Alexa Fluor 647 were prepared by incubating proteins (at concentrations of 2 mg mL$^{-1}$ in 1× PBS pH 7.4 with 1/10 volume 1 M K$_2$HPO$_4$, pH 9.0) with two-fold molar excess of Alexa Fluor 647 NHS ester (at 10 mg mL$^{-1}$ in DMSO) for 20 min at room temperature in the dark. Free dye was removed using SEC with Superdex 200 10/300 GL connected to an AKTApurifier system and equilibrated with buffer 1× PBS pH 7.4. Fractions corresponded to the expected protein pick were pulled and concentrated to a final concentration of 2 mg mL$^{-1}$ using 10000 NMWL Amicon Ultra-4 ultrafiltration devices at $4000 \times g$ and 4 °C on a Allegra X-14R centrifuge. Final protein concentrations and degrees of labeling were measured using a NanoDrop 2000 Spectrophotometer. Dye-to-protein ratios ranged from 1.0 to 1.5.

**Selection of crossreactive antibodies.** Crossreactive protein binders to human and murine ELR$^+$ CXC chemokines based on the synthetic antibody single-chain variable fragment scaffold (scFv) were isolated using yeast surface display technology[29]. The yeast-displayed synthetic antibody naïve library "G" was constructed using homologous recombination-based methods[30]. The library was designed to display the synthetic scFv variants on the surface of yeast as C-terminal fusion of the a-agglutinin Aga2 protein ($^{N}$Aga2p-scFv$^{C}$). Yeast surface display vector are based on pCT-CON backbone and includes a secretory leader sequence, a sequence encoding for the Aga2p protein, a sequence encoding for the influenza hemagglutinin epitope tag (HA; $^{N}$YPYDVPDYA$^{C}$), a 15 amino-acid flexible linker ($^{N}$GGGGSGGGGSGGGGS$^{C}$), a sequence encoding for the synthetic scFv in the light (V$_L$) to heavy (V$_H$) chain orientation, separated by another 15 amino-acid flexible linker ($^{N}$GTTAASGSSGGSSSGA$^{C}$). A sequence encoding for c-myc epitope tag (c-myc; $^{N}$EQKLISEEDLQ$^{C}$) was inserted at the C-terminus of the gene encoding the scFv to obtain $^{N}$Aga2p-HA-(G$_4$S)$_3$-V$_L$-linker-V$_H$-c-myc$^{C}$ fusion proteins. Yeast-display selection was performed by using an amount of yeast cells at least ten-fold larger than (i) the initial estimated naïve library size ($1 \times 10^9$ unique clones) or (ii) the number of cells isolated from the previous round of either magnetic bead screening or flow cytometry sorting. The yeast cells display naïve library were grown in SD-CAA medium at 30 °C with shacking (250 rpm) and surface protein expression induced in SG-CAA media for 20 h at 20 °C with shacking (250 rpm). Before positive selection, yeast populations ($1 \times 10^{10}$) underwent three sequential cycles of negative selection using uncoated Dynabeads biotin binder magnetic beads (Thermo Fisher Scientific). Ten-fold diversity library depleted of streptavidin-coated beads binders was following screened against highly diverse human (hCXCL1, hCXCL5, and hCXCL8) and murine (mCXCL1, mCXCL2, and mCXCL5) biotinylated ELR$^+$ CXC chemokines captured on magnetic beads. Two iterative cycles of magnetic bead selections followed by four cycles of fluorescence-activated cell sorting (FACS) were applied. Complex positive selection schemes, in which ten-fold of the cell output isolated from a pathway was incubated with a diverse ELR$^+$ CXC chemokine target in the following pathway, were performed to force crossreactivity and thus enhance the probabilities of isolating promiscuous protein binders. Each cycle comprises growth of yeast cells, expression of the synthetic antibodies on the surface, binding to the immobilized ELR$^+$ CXC chemokine ligands, washing and expansion of the isolated bound yeast cells. Cells were washed using ice-cold PBSA buffer (1× PBS pH 7.4 supplemented with 0.1% w/v bovine SA fraction V). For FACS, highly crossreactive protein binders were selected using a two-color labeling scheme based on fluorescent-conjugated detection reagents for expression (anti-c-myc epitope tag) and binding to ELR$^+$ CXC chemokine (anti-biotin) at recommended dilutions (Supplementary Table 12). Notably, highly avidity magnetic and fluorescently labeled reagents (e.g. streptavidin and neutravidin) saturated with diverse biotinylated ELR$^+$ CXC chemokines were used during all selection cycles. The use of highly avid reagents increase the likelihood of isolating crossreactive low-affinity binders from the naïve library by exploiting the multivalent interaction between yeast cells and the pre-loaded target. Sorting was performed on BD FACSAria I and III sorter instruments (BD Biosciences) and data evaluated using FlowJo v.10.0.7 software (Tree Star). After six cycles of iterative selections, DNA plasmid was extracted from isolated yeast cells using Zymoprep Yeast Plasmid Miniprep II Kit (Zymo Research). Extracted DNA plasmids were further amplified in E. coli, purified, and used (i) to reveal the amino-acid sequence of each selected protein binder by DNA sequencing, (ii) to transform new yeast cells to determine the binding affinity of single protein binder using yeast cell surface titrations, and (iii) as a template to prepare mutagenized DNA for further library generation and co-evolution of both binding affinity and promiscuity.

**Determination of equilibrium dissociation constants.** The equilibrium dissociation constant ($K_D$) of each individual selected protein binder towards single CXC chemokines was determined using yeast surface display titrations[29]. DNA plasmids encoding single protein binder clones were transformed into genetically modified Saccharomyces cerevisiae yeast cells (EBY100 strain) using Frozen-EZ Yeast Transformation II Kit (Zymo Research) and plated on selective SD-CAA solid agar media. Individual colonies were inoculated in 5 mL SD-SCAA cultures, grown to mid-log phase (OD$_{600}$ = 2–5) in SD-CAA media at 30 °C with shacking (250 rpm). Cells were induced in SG-CAA media for 20 h at 20 °C with shacking (250 rpm). The binding assays were conducted in 96-well plates (Corning) containing $1 \times 10^4$ induced cells per well. Non-displaying yeast cells ($1 \times 10^5$) were added to each well and mixed to induced cells to ensure (i) proper cell pelleting and (ii) an excess of soluble CXC chemokine target over total number of yeast-displayed protein binders ($5 \times 10^4$ copies of proteins per yeast cell) in solution. Yeast cells displaying protein binders were incubated with varying concentrations of soluble CXC chemokine fusions ($^{N}$CXCL-SA$^{C}$) bearing the His$_6$ tag and the primary chicken anti-c-myc epitope tag (1:1000) antibody (Gallus Immunotech) overnight at 4 °C with shaking (150 rpm). Twelve to 18 different concentrations of pure $^{N}$CXCL-SA$^{C}$ fusion proteins, ranging from 10 pM to 2 μM, were applied spanning a range of concentrations ten times both above and below the expected $K_D$ value. After primary incubation, cells were pelleted ($2500 \times g$ for 5 min at 4 °C) and washed twice with 200 μL ice-cold PBSA buffer. Secondary labeling was performed with goat anti-chicken and mouse anti-His$_6$ epitope tag antibodies conjugated to Alexa Fluor dyes at recommended dilutions (Supplementary Table 12).

Analogous approach was used to determine the non-specific polyreactivity of CK129, CK138, and CK157 antibodies towards 5 structurally related CCL chemokines and 11 structurally unrelated proteins. Primary labeling was performed using chicken anti-c-myc epitope tag (1:1000) antibody and single concentration (2 μM) of soluble proteins. Secondary labeling was performed with goat anti-chicken and either streptavidin or mouse anti-His$_6$ epitope tag antibody conjugated to Alexa Fluor dyes at recommended dilutions (Supplementary Table 12). The 96-well plates were run on a high-throughput plate sampler iQue Screener (IntelliCyt) or individually analyzed on an Accuri C6 Flow Cytometer (BD Accuri Cytometers). Data were evaluated using FlowJo v.10.0.7 software. To ensure that the difference in binding was not due to variations of number of proteins expressed on the surface of yeast cell, the median fluorescence intensity (MFI$_{BIND}$) from binding signal was normalized to the median fluorescence intensity (MFI$_{DISP}$) from display signal. The normalized (binding/display = MFI$_{BIND}$/MFI$_{DISP}$) median fluorescence intensity as a function of protein concentration was used to determine the $K_D$ values for all clones of interest. Values reported here are the results of three independent experiments and are presented as mean (dots) ± s.e.m. (bars).

**Engineering antibody affinity and crossreactivity.** Two series of random mutagenesis and FACS-based selections (named I and II) were applied to improve both the binding affinity and crossreactivity of three promiscuous clones: CK1, CK2, and CK4. Random mutagenesis libraries were generated by error-prone PCR[29]. To ensure a mutagenesis rate of approximately 1–2 amino-acid mutated residues distributed randomly throughout the entire gene, 1 ng of DNA template encoding the CK1, CK2, and CK4 binders were PCR amplified for 15 cycles using Taq DNA polymerase (New England BioLabs), analog nucleotides (2 μM 8-oxo-dGTP and 2 μM dPTP) and flanking oligonucleotide primers (forward: 5′-GGAGGCGGTAGCGGAGGCGGAGGGTCGGCTAGC-3′; reverse: 5′-GTCCTCTTCAGAAATAAGCTTTTGTTCGGAT-3′; Integrated DNA Technologies). The mutagenized PCR products were further purified, re-amplified for additional 30 cycles in the absence of analog nucleotides, and combined with SalI-HF, NheI-HF, and BamHI-HI (New England BioLabs) digested pCT-CON vector. Pre-mixed DNA linearized vector and PCR insert (1 μg μL$^{-1}$) was electroporated into freshly prepare EBY100 competent cells, where the full constructs are reassembled via homologous recombination[29]. Transformed cultures were recovered and expanded in SD-SCAA. Small portions of transformed cells were serially diluted and titrated on SD-SCAA plates to confirm the final library sizes (10$^8$ range). Library quality and diversity was further assessed by sequencing 20 colonies of each library. All clones sequenced from the mutagenized libraries were found to be in the expected format. The yeast cells display mutagenized libraries were grown in SD-CAA medium at 30 °C with shacking (250 rpm) and surface protein expression induced in SG-CAA media for 20 h at 20 °C with shacking (250 rpm). An amount of yeast cells at least ten-fold larger than the estimated mutagenized libraries size were screened against human (hCXCL1, hCXCL5, and hCXCL8) and murine (mCXCL1, mCXCL2, and mCXCL5) biotinylated ELR$^+$ CXC chemokines using equilibrium-based selection strategies. Five to six sequential cycles of FACS were applied. Each cycle comprises growth of yeast cells, expression of the binders on the surface, binding to the immobilized ELR$^+$ CXC chemokine ligands, washing, and expansion of the isolated bound yeast cells[29]. Complex selection schemes, in which ten-fold of the cell output isolated from a pathway was incubated with a diverse ELR$^+$ CXC chemokine target in the following pathway, were performed to force crossreactivity and thus enhance the probabilities of isolating promiscuous protein binders. Decreasing concentrations of biotinylated ELR$^+$ CXC chemokines up to five- and ten-fold below the measured $K_D$ were used for each round of selection in order to select for crossreactive clones with improved affinity. Secondary fluorescent-conjugated detection reagents for FACS were constantly alternated to avoid enrichments of clones that could bind to them (Supplementary Table 12). Sorting was performed on BD FACSAria I and III sorter instruments and data evaluated using FlowJo v.10.0.7 software. After multiple cycles of iterative selections, DNA plasmid was extracted from isolated yeast cells and used for further DNA sequencing and single clone characterization as described above.

**Combinatorial site-directed mutagenesis.** Individual mutations from different protein binders can be combined to further enhance affinity and crossreactivity. Toward this goal, a third step of site-directed mutagenesis (named III) was applied to combine mutations derived from different CK1 and CK2 lineage-derived clones. Site-directed mutagenesis was performed by whole plasmid PCR using Quik-Change site directed mutagenesis kit (Agilent Technologies) and pairs of complementary primers carrying single point mutations (Integrated DNA Technologies). The DNA sequences encoding CK63, CK66 and CK72 (CK1 lineage) and CK108, CK111 and CK119 (CK2 lineage) were used as templates to generate 15 (CK131-CK145) and 13 (CK146-CK158) variants, respectively, each including different combinations of CDR and FWR mutations. All constructs were verified by DNA sequencing (see Supplementary Data 4 for information about oligonucleotide primers, DNA, and amino-acid sequences). Single mutants were displayed on the surface of EBY100 cells using Frozen-EZ Yeast Transformation II Kit and plated on selective SD-CAA solid agar media. Individual colonies were inoculated in 5 mL SD-SCAA cultures, grown to mid-log phase (OD$_{600}$ = 2–5) in SD-CAA media at 30 °C with shacking (250 rpm). Cells were induced in SG-CAA

media for 20 h at 20 °C with shacking (250 rpm). The equilibrium dissociation constant ($K_D$) of each individual clone towards single CXC chemokines was determined by using yeast surface display titrations combined with flow cytometry as described above.

**Display of chemokines on the surface of yeast cells.** The ELR$^+$ and ELR$^-$ CXC chemokines were displayed on the surface of yeast as N-terminal fusion of the a-agglutinin Aga2 protein ($^N$CXCL-Aga2p$^C$). Yeast surface display vectors are based on pCT backbone[29]. Constructs for surface display of $^N$CXCL-Aga2p$^C$ fusion proteins were generated using DNA assembly methods and enzymes described above. The synthetic DNA coding for the active protein form of 12 human and murine ELR$^+$ CXC chemokines and 8 human and murine ELR$^-$ CXC chemokines were obtained from GeneArt Gene Synthesis. The de novo synthesized genes encoding for the active processed form of each CXC chemokine were subsequently inserted into a previously modified yeast-display pCT vector containing a DNA sequence encoding for a secretory leader sequence, a three amino-acid flexible spacer ($^N$GGG$^C$), a sequence encoding for c-myc epitope tag (c-myc; $^N$EQKLISEEDLQ$^C$) followed by a sequence encoding for the Aga2p protein to obtain $^N$CXCL-(G$_3$)-c-myc-Aga2p$^C$ fusion proteins. The process of the leader sequence during the secretory pathway allows for a precisely cleaved N-terminus that is crucial for the activity of the mature chemokines. Genes encoding $^N$CXCL-(G$_3$)-c-myc-Aga2p$^C$ fusion proteins were further sub-cloned into a new pCT vector via Bpu10I and XhoI (New England BioLabs) restriction enzymes except for mCXCL2 for which PstI-HF and XhoI (New England BioLabs) restriction enzymes were used. All constructs were verified by DNA sequencing and termed $^N$CXCL-Aga2p$^C$ fusion proteins (see Supplementary Data 5 for information about protein accession numbers, oligonucleotide primers, DNA, and amino-acid sequences). EBY100 cells were transformed with pCT vectors encoding $^N$CXCL-Aga2p$^C$ fusion proteins using Frozen-EZ Yeast Transformation II Kit. Cells were grown to mid-log phase in SD-CAA media at 30 °C with shacking (250 rpm) and induced in SG-CAA media for 20 h at 20 °C with shacking (250 rpm). Staining of C-terminus c-myc epitope tag indicated that all the CXC chemokines are expressed well on the surface of yeast (approximately 10$^5$ copies per cell, a standard for yeast surface display). The proper folding of yeast-displayed CXC chemokines was assessed by measuring some displayed CXC chemokines to a panel of commercial neutralizing antibodies (Supplementary Table 13). The equilibrium dissociation constant ($K_D$) of each individual yeast displayed CXC chemokine toward soluble SA129, SA138, and SA157* SA–antibody fusions was determined by using yeast surface display titrations combined to flow cytometry. The binding assays were conducted in 96-well plates containing 1 × 10$^4$ induced cells per well pre-mixed with 1 × 10$^5$ non-displaying yeast cells. Yeast cells displaying single CXC chemokine were incubated with soluble SA–antibody fusions SA129, SA138, and SA157* bearing the His$_6$ tag and the primary chicken anti-c-myc epitope tag (1:1000) antibody overnight at 4 °C with shaking (150 rpm). Twelve to 18 different concentrations of pure $^N$CXCL-SA$^C$ fusion proteins, ranging from 10 pM to 20 μM, were applied spanning a range of concentrations ten times both above and below the expected $K_D$ value. After primary incubation, cells were pelleted (2500 × g for 5 min at 4 °C) and washed twice with 200 μL of ice-cold PBSA buffer. Secondary labeling was performed with goat anti-chicken and mouse anti-His$_6$ epitope tag antibodies conjugated to Alexa Fluor dyes at recommended dilutions (Supplementary Table 12). The 96-well plates were run on an iQue Screener and data evaluated using FlowJo v.10.0.7 software. The normalized (binding/display = MFI$_{BIND}$/MFI$_{DISP}$) median fluorescence intensity as a function of SA–antibody fusions concentration was used to determine the $K_D$ values for all selected chemokines. Values reported here are the results of three independent experiments and are presented as mean (dots) ± s.e.m. (bars).

**Epitope mapping.** Structurally buried hydrophobic amino acids (I23, V40, A42, L52, V59, I62 and I63) as well as proline (P20, P31, P33, P54, and P57) and cysteine (C9, C11, C35, and C52) residues that are crucial for overall folding and stability of the chemokine were left unaltered. The wild-type hCXCL1 (hCXCL1$^{WT}$) was displayed on the surface of yeast as the amino terminus fusion of the a-agglutinin Aga2 protein ($^N$hCXCL1$^{WT}$-Aga2p$^C$). Gene encoding $^N$hCXCL1$^{WT}$-(G$_3$)-c-myc-Aga2p$^C$ fusion protein was sub-cloned into a new pCT vector via Bpu10I and XhoI (New England BioLabs) restriction enzymes. The obtained pCT-hCXCL1$^{WT}$-Aga2 vector was used as the template for the site-directed mutagenesis. Mutagenic oligonucleotides were designed to introduce single point mutations at the desired sites and generate 54 hCXCL1 variants (pCT-hCXCL1$^{ALAn}$-Aga2, $^N$hCXCL1$^{ALAn}$-Aga2p$^C$; see Supplementary Data 6 for information about hCXCL1 mutants, oligonucleotide primers, DNA, and amino-acid sequences). Binding of hCXCL1$^{WT}$ and single alanine mutants (hCXCL1$^{ALAn}$) displayed on the surface of yeast toward soluble SA129, SA138, and SA157* SA–antibody fusions was assessed by using flow cytometry. EBY100 cells were transformed with pCT vectors encoding $^N$hCXCL1$^{WT}$-Aga2p$^C$ and $^N$hCXCL1$^{ALAn}$-Aga2p$^C$ fusion proteins using Frozen-EZ Yeast Transformation II Kit. Individual colonies were inoculated in 5 mL SD-SCAA cultures, grown to mid-log phase (OD$_{600}$ = 2–5) in SD-CAA media at 30 °C with shacking (250 rpm) and induced in SG-CAA media for 20 h at 20 °C with shacking (250 rpm). The binding assays were conducted in 96-well plates containing 1 × 10$^4$ induced cells per well pre-mixed with 1 × 10$^5$ non-displaying yeast cells. The level of expression of hCXCL1$^{WT}$ and hCXCL1$^{ALAn}$ mutants displayed on the surface of yeast was assessed by staining the C-terminus c-myc epitope tag.

Yeast cells displaying hCXCL1$^{WT}$ and single hCXCL1$^{ALAn}$ were then incubated with soluble SA–antibody fusions SA129, SA138, and SA157* bearing the His$_6$ tag and the primary chicken anti-c-myc epitope tag (1:1000) antibody overnight at 4 °C with shaking (150 rpm). The binding epitopes of two commercial mouse-derived monoclonal antibodies Ab275 (clone 20326) and Ab276 (clone 31716) targeting hCXCL1 were also determined. High quality epitope maps were achieved by performing the assays at concentrations of soluble SA–antibody fusions and antibodies that were equivalent to their $K_D$ binding values for the hCXCL1$^{WT}$ (2.5 nM for SA129, 100 nM for SA138, 1.5 µM for SA157*, 0.1 nM for Ab275, and 0.25 nM for Ab276). After primary incubation, cells were pelleted ($2500 \times g$ for 5 min at 4 °C) and washed twice with 200 µL of ice-cold PBSA buffer. Secondary labeling was performed with goat anti-chicken and either mouse anti-His$_6$ epitope tag or goat anti-mouse antibodies conjugated to Alexa Fluor dyes at recommended dilutions (Supplementary Table 12). The 96-well plates were run on an iQue Screener and data evaluated using FlowJo v.10.0.7 software. To ensure that the difference in binding were not due to variations of number of proteins expressed on the surface of yeast cell, the MFI from binding signal measured for single hCXCL1$^{WT}$ and hCXCL1$^{ALAn}$ (MFI$_{BIND}$) were normalized to the MFI from display signal (MFI$_{DISP}$). The normalized (binding/display = MFI$_{BIND}$/MFI$_{DISP}$) values obtained for each hCXCL1$^{ALAn}$ variant were further normalized for the normalized value obtained for hCXCL1$^{WT}$ and plotted as (MFI$_{BIND}^{ALAn}$/MFI$_{DISP}^{ALAn}$)/ (MFI$_{BIND}^{WT}$/MFI$_{DISP}^{WT}$) providing us with a value, ranging from 0.0 to 1.0, that corresponds to the contribution of each amino-acid residues upon binding with the corresponding SA fusion or neutralizing antibody. Values reported here are the results of three independent experiments and are presented as mean (dots) ± s.e.m. (bars).

**Competitive binding assay on yeast cells.** A competitive flow cytometry-based binding assay was performed to further validate the identified hCXCL1 binding epitopes in different ELR$^+$ CXC chemokines. The assay was conducted in 96-well plates containing $1 \times 10^4$ induced cells per well pre-mixed with $1 \times 10^5$ non-displaying yeast cells. Yeast cells displaying the ELR$^+$ CXC chemokines hCXCL1, hCXCL5, hCXCL8, mCXCL1, and mCXCL2 were pre-incubated at 4 °C with concentration of soluble un-biotinylated protein SA fusions and neutralizing antibodies ("blocking reagents") that are equal to 100-times their $K_D$ values ($C_B = 100 \times K_D$). After 90 min, soluble biotinylated protein SA fusions and neutralizing antibodies ("detection reagents") were added at concentrations that are equals to their $K_D$ values ($C_D = K_D$). The incubation time was 30 min at 4 °C with shacking (150 rpm). The cells were then pelleted at $2500 \times g$ for 5 min and 4 °C on an Allegra X-14R centrifuge and washed twice with 200 µL ice-cold PBSA buffer. Secondary labeling was performed at 4 °C by using goat anti-chicken and either streptavidin or goat anti-mouse and anti-rat antibodies conjugated to Alexa Fluor 647 at recommended dilutions (Supplementary Table 12). After 30 min, the cells were pelleted at $2500 \times g$ for 5 min and 4 °C on an Allegra X-14R centrifuge and washed twice with 200 µL ice-cold PBSA buffer. The 96-well plates were run on an iQue Screener and data evaluated using FlowJo v.10.0.7 software. To ensure that the difference in binding was not due to variations of number of proteins expressed on the surface of yeast cell, the determined MFI$_{BIND}$ was normalized to the MFI$_{DISP}$. The obtained normalized binding/display (MFI$_{BIND}$/MFI$_{DISP}$) values were further normalized to the value obtained in the absence of "blocking reagent" providing us with a percentage value, ranging from 0 to 100 %, that corresponds to the residual binding observed upon blocking with the corresponding un-biotinylated SA fusion or neutralizing antibody. Values reported here are the results of two independent experiments and are presented as mean (dots) ± s.e.m. (bars).

**Competitive binding assay on mammalian cells.** The binding of two biotinylated human ELR$^+$ CXC chemokines (hCXCL1 and hCXCL8) to the human CXCR1 and CXCR2 receptors was assessed by using flow cytometry-based binding assay. Human embryonic kidney 293 (HEK293) cells that stably express the human CXCR1 (HEK293-IL8RA) and CXCR2 (HEK293-IL8RB) receptors were kindly provided by Dr. Ji Ming Wang (National Cancer Institute at Frederick, Maryland). Cell lines were not authenticated and were tested for mycoplasma contamination. Transfected HEK293 cells were maintained in Dulbecco's modified Eagle's medium (Thermo Fisher Scientific) supplemented with 10% v/v FBS (Thermo Fisher Scientific), 1% v/v penicillin–streptomycin (Thermo Fisher Scientific), and 0.8 mg mL$^{-1}$ G418 (Thermo Fisher Scientific), and grown to approximately 80% confluence in 75 cm$^2$ flasks in a humidified incubator and an atmosphere of 95% air, 5% CO$_2$ at 37 °C. Receptor expression levels were determined by flow cytometry using fluorescently labeled monoclonal antibodies against human CXCR1 and CXCR2 receptors on an Accuri C6 Flow Cytometer (Supplementary Table 13). Cells were treated with Cell Dissociation Buffer Enzyme Free PBS based buffer (Gibcon), washed twice with cold 1× PBS pH 7.4 and resuspended in cold cell binding assay (CBA) buffer (1× PBS pH 7.4 supplemented with 1% w/v BSA and 0.1% w/v NaN$_3$) to a final density of $1 \times 10^6$ cells mL$^{-1}$. Cells were then distributed (100 µL) in 96-well plates and individual wells ($1 \times 10^5$ cells each) were incubated with various concentrations of biotinylated hCXCL1 and hCXCL8 ranging from 0.03 to 300 nM. The incubation time was 30 min at 4 °C with shacking (150 rpm). The cells were then pelleted at $600 \times g$ for 5 min and 4 °C on an Allegra X-14R centrifuge and washed once with 200 µL ice-cold CBA buffer. Specific binding of biotinylated ELR$^+$ CXC chemokines to CXCR receptors was detected by incubating the cells with

Alexa Fluor 647-labeled streptavidin (1:200; Thermo Fisher Scientific) for 30 min at 4 °C with shaking. Cells were then pelleted at $600 \times g$ for 5 min and 4 °C on an Allegra X-14R centrifuge, and washed twice with 200 µL ice-cold CBA buffer. Cells were then resuspended in 50 µL ($2 \times 10^3$ cell µL$^{-1}$ final concentration) of cold CBA buffer and analyzed by flow cytometry on an iQue Screener. Data were evaluated using FlowJo v.10.0.7 software. MFI were normalized to the maximal value obtained, expressed as a percentage and plotted as a function of varying ELR$^+$ CXC chemokine concentration. The maximal effective concentrations (EC$_{50}$) were determined by fitting a sigmoidal dose–response curve on GraphPad Prism (GraphPad Software). The same assay was used to assess the ability of promiscuous SA–antibody fusions (SA129, SA138, and SA157*) and commercial neutralizing antibodies (Ab208 and Ab275, R&D Systems; Supplementary Table 13) to compete for binding of biotinylated ELR$^+$ CXC chemokines (hCXCL1 and hCXCL8) to their cognate CXCR1 and CXCR2 receptors. HEK293 cell lines expressing human CXCR1 and CXCR2 receptors were incubated with biotinylated hCXCL1, and hCXCL8 chemokines as "agonist", at final concentration equal to EC$_{50}$ values, in the presence of varying concentrations of "antagonists" (SA129, SA138, SA157*, Ab208, and Ab275), followed by staining with fluorescently labeled streptavidin. Antagonists were serially diluted in 1× PBS pH 7.4 to obtain final concentrations that cover the range from 0.3 nM to 3 µM. Concentrations ranging from 0.003 to 30 µM were used for the antagonist SA157*. Measured MFI were normalized to the maximal value obtained, expressed as a percentage and plotted as a function of varying concentrations of "antagonists". The half-maximal inhibitory concentration (IC$_{50}$) values were determined by fitting a sigmoidal dose–response curve on GraphPad Prism. The IC$_{50}$ values were further converted to inhibition constants $K_i$ by using the Cheng–Prusoff equation $K_i = IC_{50}/([L]/EC_{50} + 1)$ where [L] is the fixed concentration of "agonist" biotinylated ELR$^+$ CXC chemokine and EC$_{50}$ is the concentration of "agonist" that results in half-maximal activation of the receptor. Values reported here are the results of three independent experiments. The $K_i$ and $K_D$ values, specified in units of molar concentration (M), are converted to the p$K_i$ and p$K_D$ scale using p$K_i = -\log_{10}(K_i)$ and p$K_D = -\log_{10}(K_D)$, respectively. Higher values of p$K_i$ and p$K_D$ indicate exponentially greater potency. Data are presented as mean (dots) ± s.e.m. (bars).

**Neutrophil isolation.** Human neutrophils were purified directly from human whole blood by immunomagnetic negative selection using EasySep Direct Human Neutrophil Isolation Kit (Stemcell Technologies). Whole blood from healthy human volunteers was obtained from Research Blood Components, LLC. Murine neutrophils were isolated directly from mouse bone marrow by immunomagnetic negative selection using EasySep Mouse Neutrophils Enrichment Kit (Stemcell Technologies). The ends of femur and tibia derived from female C57BL/6 mice (Taconic) were cut and the bone marrow cells flushed using a syringe equipped with a 23-gauge needle. Cell clumps and debris were removed by gently passing the cell suspension through a 70 µm mesh nylon strainer. Both human and murine neutrophils were then pelleted at $1000 \times g$ for 5 min at 4 °C on an Allegra X-14R centrifuge, the supernatant discarded, and the cells washed by adding ice-cold PBE buffer (1× PBS pH 7.4 supplemented with 2 mM EDTA, 0.5% w/v BSA, Ca$^{2+}$ and Mg$^{2+}$ free) to obtain a final cell density of $10^6$ cells mL$^{-1}$. The washing step was repeated one time more and the washed cells resuspended at $10^7$ cells mL$^{-1}$ in ice-cold PBE buffer. Purity of human neutrophils was assessed by using APC-conjugated anti-human CD16 (clone 3G8; BioLegend), FITC-conjugated anti-human CD66b antibody (clone G10F5; BioLegend), and PE-conjugated anti-human CD45 antibody (clone HI30; BioLegend). Purity of murine neutrophils was assessed by using APC-conjugated anti-mouse CD11b (clone M1/70; BioLegend) and PE-conjugated anti-mouse Ly6G/Ly-6C (Gr-1) (clone RB6-8C5; BioLegend; Supplementary Table 13). Purified and labeled human and murine neutrophils were further used for calcium signaling experiments.

**Intracellular calcium mobilization assay.** The ability of engineered SA fusion antibody to block the capacity of human and murine ELR$^+$ CXC chemokines to signal through CXCR1 and CXCR2 receptors, resulting in an increase of the intracellular calcium concentration, was tested on both human and murine freshly purified neutrophils, respectively. Purified human and murine neutrophils in sterile ice-cold PBE buffer were loaded for 30 min at 37 °C in the dark with 2 mM cell permeable ratiometric fluorescent dye Indo-1 AM (Thermo Fisher Scientific) resuspended in 100% v/v dry DMSO to obtain a final concentration of 4 µM. Indo-1 loaded neutrophils were then pelleted at $1000 \times g$ for 5 min at 4 °C on a Allegra X-14R centrifuge, the supernatant discarded and the cells washed by adding ice-cold Cell Loading (CL) buffer (1× HBSS, pH 7.4, 0.5% w/v BSA, 1 mM Ca$^{2+}$ and 1 mM Mg$^{2+}$) to obtain a final cell density of $10^7$ cells mL$^{-1}$. The washing step was repeated one time more and the washed cells were resuspended at $5 \times 10^6$ cells mL$^{-1}$ in ice-cold CL buffer. Aliquots of $10^6$ cells per tube (200 µL) were prepared, individually pre-warmed at 37 °C for 10 min, and stimulated with varying concentrations of "agonist" ELR$^+$ CXC chemokines ranging from 0.03 to 300 nM. Samples were analyzed on a BD LSR II flow cytometer (BD Biosciences). Intracellular calcium levels were measured at 405/30 nm (Indo-1 low) and 485/20 nm (Indo-1 high) emission fluorescence after excitation at 355 nm. Baseline fluorescence was recorded for 60 s before the addition of "agonist" ELR$^+$ CXC chemokines and fluorescence measured for an additional 240 s. The MFI at 405/30 and 485/20 nm were recorded, the ratio of two wavelengths calculated (Indo-1 ratio)

and plotted as a function of time (seconds). Area under the curve (AUC), calculated as an integral over time, was determined using FlowJo v.10.0.7 software. The obtained values were normalized to the maximal response acquired, expressed as the percentage of activity. The $EC_{50}$ was determined by fitting a sigmoidal dose–response curve on GraphPad Prism. The same assay was used to assess the ability of "antagonist" SA–antibody fusions SA129, SA138, and SA157* to antagonize the $ELR^+$ CXC chemokine-mediated receptors activation and downstream intracellular calcium mobilization. Commercial neutralizing antibodies targeting human hCXCL1 (Ab275), hCXCL5 (Ab654), hCXCL8 (Ab208), and murine mCXCL1 (Ab453) and mCXCL2 (Ab452) were included as positive controls (Supplementary Table 13). Indo-1-loaded neutrophils were incubated with hCXCL1, hCXCL5, hCXCL8, mCXCL1, and mCXCL2 chemokines as "agonist", at final concentration equal to $EC_{50}$ values, in the presence of varying concentrations of "antagonist" SA–antibody fusions and neutralizing antibodies. Antagonists were serially diluted in ice-cold CL buffer to obtain final concentrations that cover the range from 10 pM to 10 μM. Intracellular calcium levels were measured as described above. The obtained values were normalized to the maximal response acquired and expressed as the percentage of activity plotted as a function of varying concentrations of "antagonists". Values reported here are the results of three independent experiments. Data are presented as mean (dots) ± s.e.m. (bars). The $IC_{50}$ values were determined by fitting a sigmoidal dose–response curve on GraphPad Prism. The $IC_{50}$ values were further converted to inhibition constants $K_i$ by using the Cheng–Prusoff equation. The $pK_i$ and $pK_D$ values were determined as described above.

**Mice.** Female C57BL/6 mice (strain number B6NTac) were purchased from Taconic. Male C57BL/6J mice (strain number 000664) and NOD/ShiLtJ mice (strain number 001976) were purchased from Jackson Laboratory. Mice carrying the KRN T-cell receptor transgene on the C57BL/6 genetic background were mated with NOD mice (Jackson Laboratory) generated and maintained in our laboratories to obtain transgene-positive arthritic K/BxN mice. Male and female 6–8 weeks mice were used in all experiments. All mice were housed in specific pathogen-free facilities at the Massachusetts Institute of Technology and Massachusetts General Hospital and the studies carried out according to the federal, state local, and ethical regulations. All animal studies were performed under protocols approved by the Committee on Animal Care (CAC) of the Massachusetts Institute of Technology and the Center for Comparative Medicine (CCM) of the Massachusetts General Hospital. No statistical methods were used to predetermine sample size. No animals were excluded from analysis.

**Pharmacokinetic studies in mice.** Female C57BL/6 mice were maintained under specific pathogen-free conditions and used at 6–8 weeks of age. A single dose (1 mg) of each Alexa Fluor 647-labeled $^NSA$-scFv$^C$ fusions (2 mg mL$^{-1}$) were injected intraperitoneally (i.p.) into three mice. At various time points (immediately after injection and at 0.5, 1, 2, 3, 5, 8, 24, 48, 72, 96, 120, 168 h post injection) blood was collected into heparin-coated capillary tubes (VWR International) and stored at 4 °C in the dark until sample collection was complete. Plasma was obtained after centrifugation ($900 \times g$ for 5 min) and transferred to new capillary tubes. Standard samples were diluted in plasma collected from untreated mice. Serial dilutions (100 μL per well) of the standards (ranging from 0.3 to 300 pg μL$^{-1}$) and plasma samples were prepared. Protein fusion concentration was determined by measurement of fluorescent intensity using a Typhoon imager (GE Healthcare) after degree of labeling correction. Fluorescence intensity was quantified using ImageJ software (NIH). To calculate $^NSA$-scFv$^C$ half-lives, fluorescent measurements were quantified by normalization to a standard curve for each antibody. Starting at the max concentration time point (3 h for all cases), pharmacokinetic profiles were fit in Graphpad Prism using a two phase non-compartmental model of the following format: $MFI(t) = Ae^{-\alpha t} + Be^{-\beta t}$, where $A$, $B$, $\alpha$, and $\beta$ represent the systemic clearance rates of a given fusion protein (Supplementary Table 11). Fast and slow half-lives $t_{1/2,\alpha}$ and $t_{1/2,\beta}$ were calculated as $\ln(2)/\alpha$ and $\ln(2)/\beta$, respectively. The total clearance (CL) was calculated by dividing the total dose by the AUC from 0 to infinity. Fits for the three mice in each group were averaged to obtain a single pharmacokinetic curve for each $^NSA$-scFv$^C$ fusion, from which total clearance rate and standard error were calculated. Values reported here are the results of triplicate and data are presented as mean (dots) ± s.e.m. (bars).

**Arthritis induction and treatment.** The K/BxN serum transfer model of inflammatory arthritis mice was used[35]. Mice carrying the KRN T-cell receptor transgene on the C57BL/6 genetic background were mated with NOD/ShiLtJ mice to obtain transgene-positive arthritic K/BxN mice. The presence of the transgene was determined by allele-specific PCR and confirmed by phenotypic assessment. Experimental arthritis was induced in recipient male C57BL/6J by transferring arthritogenic serum containing autoantibodies to the ubiquitous anti-glucose 6-phosphate isomerase (GPI) protein from transgenic 8- to 10-week-old K/BxN mice to healthy C57BL/6J resulting in synovial pannus formation and both bone and cartilage erosions that mimics the disease that develop spontaneously in transgenic mice[66]. Arthritogenic K/BxN serum (150 μL) was injected intraperitoneally (i.p.) using 26-gauge needle syringe on days 0 and 2 on healthy wild-type C57BL/6J mice and disease progress was monitored every other day for 2 weeks. For the

preventative treatment experiments, SA fusions (1 mg per mouse per day) or PBS (0.5 mL per mouse per day) were injected i.p. starting on day 0 and then treated for a total of 14 continuative days. For therapeutic treatment, mice were placed into four experimental groups so that each group had the same overall clinical score and treated with SA fusions (1 mg per mouse per day) or PBS (0.5 mL per mouse per day) for a total of 10 days. For both preventative and therapeutic experiments group size was $n = 10$. Paw thickness and clinical scores were determined every other day[66]. The clinical arthritis score was calculated for each mouse by summing the scores for the four paws: 0 = normal; 1 = erythema and swelling of one digit; 2 = erythema and swelling of two digits or erythema and swelling of ankle joint; 3 = erythema and swelling of more than three digits or swelling of two digits and ankle joint; 4 = erythema and severe swelling of the ankle, foot and digits with deformity. Visible clinical signs were scored blinded for the origin and treatment of the mice. Because different batches of serum with different potency have been used in different experiments, the measured clinical score values of each experiment were normalized to the mean of the maximum clinical score value obtained in each independent experiment in mice treated with PBS and expressed as a percentage (clinical score—% of max). Values reported here are the results of two independent experiments and are presented as mean (dots) ± s.e.m. (bars).

**Quantification of neutrophils in synovial fluid.** The number of neutrophils that accumulated in the synovial fluid were determined using flow cytometry[66]. Synovial fluid was obtained from ankle joints of 8- to 10-weeks old C57BL/6J mice on day 8 after K/BxN serum injection for all groups. Retrieved synovial fluid cells were resuspended in sterile 1% v/v FCS/PBS to obtain a final concentration of $1 \times 10^4$ cells μL$^{-1}$. For flow cytometry analysis, cells were incubated with anti-FcγRIII/II antibody (clone 2.4G2; BD Bioscience) and stained with APC-conjugated anti-murine Ly6G antibody (clone 1A8; BioLegend; Supplementary Table 13). Flow cytometry was performed with BD LSRFortessa (BD Bioscience) and analyzed with FlowJo v.10.0.7 software. Neutrophils were identified as Ly6G-positive cells in the granulocyte gate of forward and side scatter plots. Values reported here are the results of triplicate and are presented as mean ± s.e.m. (bars).

**Histology analysis.** Preventative treated mice ($n = 3$ per group) were sacrificed at day 8 after K/BxN serum injection and paws collected for histology[66]. Paws were fixed in 4% v/v formalin solution overnight and decalcified by treatment with 20% w/v EDTA solution for 2 weeks. Samples were then washed with H$_2$O mQ for at least 10 min and embedded in paraffin. Sections of 4 μm thickness were stained with hematoxylin and eosin (H&E) staining kit (Wako Pure Chemical Industries), mounted by using Mount-Quick mounting medium (Daido Sangyo Co.) and examined by light microscopy. Histopathological scoring was performed on H&E-stained ankle sections by evaluating both inflammatory cell infiltration and pannus formation as follows. Inflammatory cell infiltration: 0 = no change, 1 = focal inflammatory cell infiltration, 2 = severe and diffuse inflammatory cell infiltration. Pannus formation: 0 = no change, 1 = pannus formation at one site, 2 = pannus formation at two sites, 3 = pannus formation at more than three sites. The score of inflammatory cell infiltration and pannus formation were summed to determine a total histopathological score. The values are presented as mean ± s.e.m. (bars). For immunofluorescence staining with von Willebrand factor (vWF), paraffin-embedded tissue sections (4 μm thick) from preventative treated mice ($n = 3$ per group) were blocked with protein block (DakoCytomation) for 15 min and stained with rabbit anti-vWF polyclonal antibody (8 μg mL$^{-1}$; LifeSpan Biosciences) or normal rabbit IgG (Bioss Antibodies) as an isotype control for 60 min at 25 °C. Alexa Fluor 647-conjugated goat anti-rabbit IgG antibodies (Abcam) was used as secondary antibody and incubated for 1 h at 25 °C. The slides were examined using a fluorescence microscope (Carl Zeiss).

**Protein structure homology modeling.** The protein structure homology models of selected yeast-displayed antibody single-chain variable fragments CK129, CK138, and CK157 have been generated by using protein structure modeling program MODELLER[67] and the three-dimensional structure of a highly homolog synthetic antibody fragment as a template (PDB ID: 2KH2). Protein structures and models were rendered using PyMOL (PyMOL Molecular Graphics System, Version 1.8 Schrödinger, LLC).

**Statistical analyses.** All data are presented as mean ± s.e.m. (bars). Statistical comparisons were made between each group using one-way analysis of variance (ANOVA), followed by Tukey's test to calculate $P$-values: *$P < 0.05$, **$P < 0.01$, ***$P < 0.001$; ****$P < 0.0001$. ns: non-significant. Statistical analyses were performed using GraphPad Prism (GraphPad Software).

**Data availability.** The data that support the findings of this study are available from the corresponding author upon request.

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

## Acknowledgements

We thank Dr. Alice Tzeng and Dr. Giulia Pasqual for critical reading of this manuscript, and the staff of the Swanson Biotechnology Center at the Koch Institute for Integrative Cancer Research at MIT for technical assistance. We are grateful to Prof. B. Pentelute and Dr. A. Rabideau of the Department of Chemistry at MIT, and Eric Spooner of the Proteomics Core Facility at the Whitehead Institute for Biomedical Research for help with mass spectrometry experiments and analysis. The financial contribution from the Swiss National Science Foundation (SNSF) Fellowship for Advanced Researchers to A.A. (PP00P3_123524/1), Ludwig Cancer Research Center Postdoctoral Fellowship to A.A., Mallinckrodt Pharmaceutical Research Fellowship Award in Rheumatology Research to Y.M., Pfizer ASPIRE Rheumatology Award to Y.M., Japan Rheumatism Foundation Research Grant to Y.M., National Science Foundation (NSF) Graduate Research Fellowship to B.H.K., National Institute of General Medical Sciences (NIGMS) Inter-departmental Biotechnology Training Program at the National Institutes of Health Graduate Research Fellowship to R.L.K. (T32 GM008334-25), Rheumatology Research Foundation to A.D.L., and Global Research Outreach (GRO) program of Samsung Electronics, Co., Ltd. to K.D.W. are gratefully acknowledged.

## Author contribution

A.A. and K.D.W. conceived the study; A.A. performed the experiments and analyzed the data; Y.M. performed the K/BxN mouse experiments and analyzed the data; D.N. assisted with the protein engineering and selection experiments; B.H.K. assisted with protein production experiments; C.M. assisted with the K/BxN mouse experiments; R.L.K. assisted with the pharmacokinetic experiments; M.N.J assisted with the epitope mapping experiments; A.A., A.D.L., and K.D.W. interpreted the data and wrote the manuscript.

## Additional information

**Competing interests:** A.A., A.D.L., and K.D.W. declare that they are named on a provisional patent application 62/546814 entitled "Multiple Specificity Binders of CXC Chemokines and Uses Thereof" that has been filed in the United States Patent and Trademark Office on behalf of the Massachusetts Institute of Technology and the Massachusetts General Hospital. The remaining authors declare no competing interests.

