## [Peer Review File · Nature Communications]

Reviewers' comments:

Reviewer #1 (Remarks to the Author):

This is a really thorough and comprehensive study by Angelini and colleagues developing and testing anti-CXC chemokine antibodies. The redundancy of the chemokine system is difficult to deal with from the therapeutic point of view and the others study a system to generate diversity in antibody repertoire in an attempt to generate broad-acting antibodies. They eventually manage to develop a few antibodies with broad specificity for the CXC chemokine of choice, study antibody-epitope binding and test these in vivo. Antibodies raised worked against both the human and murine proteins.

I do not really like the title. Perhaps "Directed evolution of broadly crossreactive chemokine-blocking antibodies efficacious in murine model of arthritis." Rheumatoid arthritis (RA) is a very chronic human disease, whose aspects are only partially modeled in mice. The model chosen is reasonable it is far-fetched to say this is RA.

To me there are two major issues. First, I take the point of the authors that " this is the first systematic study reporting the selection strategy for the in vitro directed evolution of binders with such extensive promiscuity towards a panel of structurally related, yet sequence-diverse, protein targets.". This is the major message and it is a clear methodological advance with good results. I am less convinced this is much better than available tools for ligands for CXCR1/2 receptors. Yes, I agree that the chemokine system is probably an excellent system to test the proof of concept of developing broad-acting antibodies. And choosing neutrophil-active CXC chemokines is also reasonable. The authors acknowledge this and mention that current available "antagonists have shown only limited therapeutic effects. Failures of such receptor-based therapies have often been attributed to (i) differences between the orthologous rodent (pre-clinical) and human (clinical) systems and (ii) the extremely high doses of antagonist required to guarantee continuous receptor occupancy, such that all receptors in the body are antagonized". Point (ii) may be a problem for the developed antibodies as acknowledged by the authors ("To ensure complete inhibition of all ELR+ CXC chemokines present in circulation, we used fairly high doses of our engineered antibody fusions (50 mg/kg)"

I must be said there are excellent CXCR1/2 antagonists that function against the human, murine and rat receptors. These molecules have been found to be excellent at blocking neutrophil influx in vivo. Authors cite three of such references (7-9). Some of the available antagonists are allosteric inhibitors and do not necessarily suffer from the need to be used at very high doses. In the summary, authors conclude that "The present study demonstrates that simultaneous blocking of multiple CXCR2 ligands is a promising strategy to treat rheumatoid arthritis, with general implications for the development of novel therapeutics for multifactorial diseases." Blocking CXCR2 is already known to be promising for rheumatoid arthritis (not fulfilled). It may be that broad-acting antibodies will be better. I find the abstract does little to help to "sell" the major point of the paper (mentioned at the beginning of the paragraph). The discussion is way better.

Another major point I had refers to the testing of the best final antibodies against other chemokines. At least to evaluate whether enhancing effects against other chemokines would not be accompanied by non-specific binding to most chemokines - specially because they could map some antibody binding to carbohydrate binding sites. It is mentioned that " Importantly, this was not merely due to non-specific polyreactivity of our engineered antibodies, as no binding was detected toward a panel of unrelated proteins (Supplementary Fig. 7e)." I cannot see why other chemokines and even other non-chemokine chemoattractants (e.g.. C5a) have not been used as irrelevant protein.

Reviewer #2 (Remarks to the Author):

The paper describes the results of a colossal effort, in trying to generate broadly-crossreactive antibodies, capable of simultaneous recognition of multiple (ELR+)CXC chemokines.

The goal to isolate one antibody clone, which could recognize with high affinity ALL (ELR+)CXC chemokines was not reached. However, two attractive clones (SA129 and SA138), which exhibited cross-reactivity with certain chemokine targets, were isolated and characterized in detail.

One of these antibodies (SA138), used as scFv-albumin fusion, inhibited the activity of certain chemokines and provided an inhibition of arthritis progression in a murine model of the disease, which features a transient elevation of clinical scores.

The paper is exemplary in the thoroughness of the approach that was taken, as well as in the quality and completeness of documentation. Even though "total" cross-reactivity was not achieved, one of the clones exhibited encouraging biological activity. Most importantly, the paper highlights the potential and the limitations associated with the desire to generate cross-reacting antibodies against homologous protein targets, using a combination of protein evolution and yeast display techniques.

I recommend that this article is accepted for publication. The only request that I have (and that I strongly believe should be implemented in the article) is that the authors "reduce" the tone of some statements and provide a more balanced evaluation of the results that they have achieved, as well as of the challenges that still remain.

The therapeutic results, while promising, are not spectacular. In an ideal world, I would have liked to see SA138 tested on a second model of arthritis (e.g., collagen-induced arthritis), but I understand that already a lot of work had gone into this article.

Moreover, while the wish to engineer cross-reactive antibodies remains important, there are clear limitations (which are also evident from the results of this very extensive discovery program).

In other words, the methodology and the results obtained (both the positive and the negative ones) are the most valuable aspects of the paper, not necessarily individual "wow factors".

Reviewer #3 (Remarks to the Author):

There is also unmet need for the treatment of arthritides including rheumatoid arthritis (RA). The role of chemokines and chemokine receptors have been widely characterized in the pathogenesis of RA. Yet, almost all studies using antibodies or small molecule inhibitors of chemokines or chemokine receptors failed. Reasons for failure may include cross-species incompatibility, processing of the targeting molecule, wrong target and timing, receptor occupancy, etc. Until now, CC chemokines were mostly chosen as targets, several CCR1, CCR2 and CCR5 antagonists were developed but all failed in phase I-II trials due to reasons described above. Therefore one could come to the conclusion that anti-chemokine/receptor targeting using conventional antibodies or small molecule inhibitors is a "no-go" in RA.

The present study utilizes a significantly different approach: using single-chain antibodies with multiple chemokine specificities. This is an absolutely original approach, no published data have become available using this approach in the treatment of arthritis. Authors picked two albumin-bound antibodies, SA129 with specificity for CXCL1 and SA138 with multiple specificities for CXCL1,2,3,5. All these chemokines bind to the CXCR2 receptor and CXCR2 has been characterized as the most important CXC receptor in RA binding gro- α , ENA-78, IL-18. In addition, these 3 chemokines, also identified in the present study, seem to be the most relevant in RA (maybe authors should cite the 3 original papers that defined the role of gro- α , ENA-78 and IL-8 in RA: Koch et al, J Immunol 1995; Koch et al J Clin Invest 1994; Koch et al J Immunol 1991).

The study is focused, aims are clear. Methodology is extensive, using relevant modern techniques. The K/BxN model is relevant for human RA.

The most important results (in addition to development and molecular characterization of the antibodies) are: 1. The developed antibodies successfully inhibit neutrophil ingress into the synovium. 2) These molecules can both prevent and treat murine arthritis. 3) The multi-specific

SA138 is a lot more effective in all aspects than the mono-specific SA129.

These are my remarks:

1. There has been only one study in adjuvant-induced arthritis using CXCR2 blockade. (Barsante et al, Br J Pharmacol 2008, needs to be cited). This study was very successful in animals, yet CXCR2 blockade did not evolve as a target in human RA. What could be the reason that chemokine targeting successful in animals would not work in humans? Why would the approach of the present study be any different? Do authors propose that their approach would be more successful in human RA?
2. In general, targeted therapies with broader specificity may have significantly more side effects. Did the authors observe any complications in the animals? Would they propose more side effects when using SA138 compared to SA129 due to the fact that CXCL2, CXCL5 blocked by the latter but not the former may have important role in host defense?
3. The ELR motif has been implicated in angiogenesis. Authors target ELR-positive CXC chemokines. So when they scored the grade of inflammation, could they also score vascularity of the synovial tissue?
4. What was the rationale of measuring ankle thickness rather than paw volume or ankle circumference?
5. Authors write "We next quantified the number of synovial fluid neutrophils isolated from the arthritic joints of mice treated with SA129, SA138, and SACTR antibody fusions. Synovial tissues were harvested at the peak of the disease (day 8 after disease initiation). We discovered that..." I am a bit confused. Did they actually assess synovial fluid neutrophils or synovial tissue cells? Certainly, in a chemokine study, the quantification of synovial tissue cells would be more relevant.
6. Finally, taking together their data and previous failures, would the authors summarize why their approach would be more successful and would lack all issues that led to failure of previous studies?

We thank the reviewers for their positive comments regarding our manuscript, and for pointing out that our study is “really thorough and comprehensive” (rev #1), a “colossal effort” that is “exemplary in the thoroughness of the approach that was taken, as well as in the quality and completeness of documentation” (rev #2), and represent “an absolutely original approach” (rev #3).

At the same time, the reviewers make important points, which we addressed by addition of a number of experiments to the revised version (major changes in the main text are marked in blue font).

Referees' comments:

Referee #1 (Remarks to the Author):

This is a really thorough and comprehensive study by Angelini and colleagues developing and testing anti-CXC chemokine antibodies. The redundancy of the chemokine system is difficult to deal with from the therapeutic point of view and the others study a system to generate diversity in antibody repertoire in an attempt to generate broad-acting antibodies. They eventually manage to develop a few antibodies with broad specificity for the CXC chemokine of choice, study antibody-epitope binding and test these in vivo. Antibodies raised worked against both the human and murine proteins.

Authors' reply: We thank the reviewer for the positive comments and for his/her appreciation of our approach.

I do not really like the title. Perhaps "Directed evolution of broadly crossreactive chemokine-blocking antibodies efficacious in murine model of arthritis." Rheumatoid arthritis (RA) is a very chronic human disease, whose aspects are only partially modeled in mice. The model chosen is reasonable it is far-fetched to say this is RA.

Authors' reply: We agree with the reviewer's comment that rheumatoid arthritis is only partially modeled by existing mouse models, including the one used in this work. We changed the title as suggested, and also edited the abstract.

To me there are two major issues. First, I take the point of the authors that " this is the first systematic study reporting the selection strategy for the in vitro directed evolution of binders with such extensive promiscuity towards a panel of structurally related, yet sequence-diverse, protein targets." This is the major message and it is a clear methodological advance with good results. I am less convinced this is much better than available tools for ligands for CXCR1/2 receptors. Yes, I agree that the chemokine system is probably an excellent system to test the proof of concept of developing broad-acting antibodies. And choosing neutrophil-active CXC chemokines is also reasonable. The authors acknowledge this and mention that current available "antagonists have shown

only limited therapeutic effects. Failures of such receptor-based therapies have often been attributed to (i) differences between the orthologous rodent (pre-clinical) and human (clinical) systems and (ii) the extremely high doses of antagonist required to guarantee continuous receptor occupancy, such that all receptors in the body are antagonized". Point (ii) may be a problem for the developed antibodies as acknowledged by the authors ("To ensure complete inhibition of all ELR+ CXC chemokines present in circulation, we used fairly high doses of our engineered antibody fusions (50 mg/kg)"

I must be said there are excellent CXCR1/2 antagonists that function against the human, murine and rat receptors. These molecules have been found to be excellent at blocking neutrophil influx in vivo. Authors cite three of such references (7-9). Some of the available antagonists are allosteric inhibitors and do not necessarily suffer from the need to be used at very high doses.

Authors' reply: We thank the reviewer for identifying this imprecise statement. We agree with the reviewer on the existence of non-competitive allosteric modulators of CXCR1 and CXCR2 receptors showing advantages over conventional molecules and therapeutic efficacy in both preclinical and clinical settings. We modified the introduction accordingly and added the following references:

- Bertini, R. et al. Noncompetitive allosteric inhibitors of the inflammatory chemokine receptors CXCR1 and CXCR2: prevention of reperfusion injury. *Proc Natl Acad Sci U S A* **101**, 11791-11796 (2004).
- Moriconi, A. et al. Design of noncompetitive interleukin-8 inhibitors acting on CXCR1 and CXCR2. *J Med Chem* **50**, 3984-4002 (2007).
- Allegretti, M., Cesta, M.C. & Locati, M. Allosteric Modulation of Chemoattractant Receptors. *Front Immunol* **7**, 170 (2016).

Although the CXCR1 and CXCR2 allosteric inhibitors appear to provide numerous advantages over conventional drug formats, we believe that our study presents an alternative approach to the existing CXCR1 and CXCR2 receptor-based therapies.

In the summary, authors conclude that "The present study demonstrates that simultaneous blocking of multiple CXCR2 ligands is a promising strategy to treat rheumatoid arthritis, with general implications for the development of novel therapeutics for multifactorial diseases." Blocking CXCR2 is already known to be promising for rheumatoid arthritis (not fulfilled). It may be that broad-acting antibodies will be better. I find the abstract does little to help to "sell" the major point of the paper (mentioned at the beginning of the paragraph). The discussion is way better.

Authors' reply: We agree with the reviewer that the therapeutic potential of CXCR2 blockade is not the major conclusion of the data presented in our manuscript. We

modified the abstract to emphasize as the major finding the development of a novel approach capable of targeting multiple soluble factors with a single broadly crossreactive molecule.

Another major point I had refers to the testing of the best final antibodies against other chemokines. At least to evaluate whether enhancing effects against other chemokines would not be accompanied by non-specific binding to most chemokines - specially because they could map some antibody binding to carbohydrate binding sites. It is mentioned that " Importantly, this was not merely due to non-specific polyreactivity of our engineered antibodies, as no binding was detected toward a panel of unrelated proteins (Supplementary Fig. 7e)." I cannot see why other chemokines and even other non-chemokine chemoattractants (e.g.. C5a) have not been used as irrelevant protein.

Authors' reply: We are grateful to the reviewer for raising this important point, which prompted us to carry out an experiment that was missing from our original submission. We have now measured the binding of our engineered CK129, CK138 and CK157 single-chain variable antibody fragments toward twenty soluble CXCL-SA fusions, five structurally related CCL chemokines (CCL2, CCL5, CCL20, CCL22, CCL28), and eleven structurally unrelated proteins, including the murine complement component C5a. The new data are shown below and have been also included in **Supplementary Fig. 7e** of the revised manuscript and commented on page 9.

Columns graph reporting the fluorescence binding intensities of yeast-displayed CK138 (blue), CK157 (dark gray) and CK129 (red) against twenty soluble CXCL-SA fusions, five structurally related CCL

chemokines (white; CCL2, CCL5, CCL20, CCL22, CCL28; **Supplementary Table 13**), and eleven structurally unrelated proteins (light gray; C5a: complement component C5a; TLR: toll-like receptor; FcRn: neonatal Fc receptor; Lys: lysozyme; MBP: maltose binding protein; SUMO: small ubiquitin-like modifier protein; PFO: perfringolysin O toxin; IL-2: interleukin-2; SA: serum albumin; IgG: immunoglobulin; Strep: streptavidin). Fluorescence binding signals were assessed by a flow cytometry-based assay using single concentrations (2 μ M) of soluble proteins. Obtained median values were normalized to the display median fluorescence intensities of each single yeast surface displayed partner (binding/display). Normalized values (*y*-axis) represent the means \pm SE (bars) of at least three independent experiments.

The five CCL chemokines were chosen based on *i*) their ability to bind different CCR receptors (CCL2 binds CCR2; CCL5 binds CCR1, CCR3, and CCR5; CCL20 binds CCR6; CCL22 binds CCR4; CCL28 binds CCR10) and *ii*) their diverse amino acid sequences (selected CCLs share 20-35% sequence identity among them and 12-26% sequence identity with ELR+ CXC chemokines). The experiment showed that the binding of yeast-displayed CK129 and CK138 towards CCLs and C5a is weak and comparable to that observed with structurally unrelated proteins. Contrariwise, yeast-displayed CK157 exhibits crossreactivity toward three CCL chemokines (CCL20, CCL22, CCL28) and C5a. This last result is not surprising because, although chemokines can be very different in linear sequence, they share a conserved three-dimensional fold. Moreover, all CC and CXC chemokines as well as C5a ($pI = 9.7$) are highly positively charged proteins that could potentially interact with the negatively charged patches present on the surface of CK157 (see **Supplementary Fig. 13**). This hypothesis appears to be supported by our epitope mapping experiments and tridimensional structure models that seem to confirm that the binding mode of CK157 could be similar to that of naturally evolved broadly crossreactive viral chemokine binding proteins (vCKBPs)¹⁻⁶, where a large negatively charged area engages conserved positively charged glycosaminoglycans (GAG)-binding regions located on the surface of CC and CXC chemokines, thus explaining their extensive promiscuity. Nonetheless, it is also relevant to underscore that some chemokines (CXCL7, CXCL8, CXCL9, CCL2 and CCL5) and ten structurally unrelated proteins are not recognized by CK157, implying a unique mode of binding and still restrained crossreactivity. Further studies will be required to fully understand the underlying molecular basis of CK157 binding promiscuity, an effort that is currently ongoing in the laboratory, but which we would reason is beyond the scope of a first description of these molecules.

We again thank the reviewer for the positive comments and for his/her appreciation of our work.

Referee #2 (Remarks to the Author):

The paper describes the results of a colossal effort, in trying to generate broadly-crossreactive antibodies, capable of simultaneous recognition of multiple (ELR+) CXC chemokines.

The goal to isolate one antibody clone, which could recognize with high affinity ALL (ELR+)CXC chemokines was not reached. However, two attractive clones (SA129 and SA138), which exhibited cross-reactivity with certain chemokine targets, were isolated and characterized in detail.

One of these antibodies (SA138), used as scFv-albumin fusion, inhibited the activity of certain chemokines and provided an inhibition of arthritis progression in a murine model of the disease, which features a transient elevation of clinical scores.

The paper is exemplary in the thoroughness of the approach that was taken, as well as in the quality and completeness of documentation. Even though "total" cross-reactivity was not achieved, one of the clones exhibited encouraging biological activity. Most importantly, the paper highlights the potential and the limitations associated with the desire to generate cross-reacting antibodies against homologous protein targets, using a combination of protein evolution and yeast display techniques.

I recommend that this article is accepted for publication. The only request that I have (and that I strongly believe should be implemented in the article) is that the authors "reduce" the tone of some statements and provide a more balanced evaluation of the results that they have achieved, as well as of the challenges that still remain.

The therapeutic results, while promising, are not spectacular. In an ideal world, I would have liked to see SA138 tested on a second model of arthritis (e.g., collagen-induced arthritis), but I understand that already a lot of work had gone into this article.

Moreover, while the wish to engineer cross-reactive antibodies remains important, there are clear limitations (which are also evident from the results of this very extensive discovery program).

In other words, the methodology and the results obtained (both the positive and the negative ones) are the most valuable aspects of the paper, not necessarily individual "wow factors".

Authors' reply: We thank the reviewer for the positive comments and for his/her appreciation of our approach. We agree with the Reviewer's suggestion to reduce the tone of some statements and tried to provide a more balanced evaluation of our results. We added a paragraph in the discussion section highlighting the limitations of our approach, and we state clearly the need of additional testing in other animal models of arthritis to better evaluate the therapeutic potential of our approach (page 15-16). Moreover, we

edited several sentences throughout the text.

We again thank the reviewer for the positive comments and for his/her appreciation of our work.

Referee #3 (Remarks to the Author):

There is also unmet need for the treatment of arthritides including rheumatoid arthritis (RA). The role of chemokines and chemokine receptors have been widely characterized in the pathogenesis of RA. Yet, almost all studies using antibodies or small molecule inhibitors of chemokines or chemokine receptors failed. Reasons for failure may include cross-species incompatibility, processing of the targeting molecule, wrong target and timing, receptor occupancy, etc. Until now, CC chemokines were mostly chosen as targets, several CCR1, CCR2 and CCR5 antagonists were developed but all failed in phase I-II trials due to reasons described above. Therefore one could come to the conclusion that anti-chemokine/receptor targeting using conventional antibodies or small molecule inhibitors is a "no-go" in RA.

The present study utilizes a significantly different approach: using single-chain antibodies with multiple chemokine specificities. This is an absolutely original approach, no published data have become available using this approach in the treatment of arthritis. Authors picked two albumin-bound antibodies, SA129 with specificity for CXCL1 and SA138 with multiple specificities for CXCL1,2,3,5. All these chemokines bind to the CXCR2 receptor and CXCR2 has been characterized as the most important CXC receptor in RA binding gro- α , ENA-78, IL-18. In addition, these 3 chemokines, also identified in the present study, seem to be the most relevant in RA (maybe authors should cite the 3 original papers that defined the role of gro- α , ENA-78 and IL-8 in RA: Koch et al, *J Immunol* 1995; Koch et al *J Clin Invest* 1994; Koch et al *J Immunol* 1991).

The study is focused, aims are clear. Methodology is extensive, using relevant modern techniques. The K/BxN model is relevant for human RA.

The most important results (in addition to development and molecular characterization of the antibodies) are: 1. The developed antibodies successfully inhibit neutrophil ingress into the synovium. 2) These molecules can both prevent and treat murine arthritis. 3) The multi-specific SA138 is a lot more effective in all aspects than the mono-specific SA129.

Authors' reply: We thank the reviewer for the overall positive comments and for suggesting some original papers that we have now included in the revised manuscript.

- Koch, A.E. et al. Growth-related gene product alpha. A chemotactic cytokine for neutrophils in rheumatoid arthritis. *J Immunol* 155, 3660-3666 (1995).

- Koch, A.E. et al. Epithelial neutrophil activating peptide-78: a novel chemotactic cytokine for neutrophils in arthritis. *J Clin Invest* **94**, 1012-1018 (1994).
- Koch, A.E. et al. Synovial tissue macrophage as a source of the chemotactic cytokine IL-8. *J Immunol* **147**, 2187-2195 (1991).

These are my remarks:

1. There has been only one study in adjuvant-induced arthritis using CXCR2 blockade. (Barsante et al, *Br J Pharmacol* 2008, needs to be cited). This study was very successful in animals, yet CXCR2 blockade did not evolve as a target in human RA. What could be the reason that chemokine targeting successful in animals would not work in humans? Why would the approach of the present study be any different? Do authors propose that their approach would be more successful in human RA?

Authors' reply: We thank the Reviewer for pointing out this reference, which has now been added to the revised manuscript.

- Barsante, M.M. et al. Blockade of the chemokine receptor CXCR2 ameliorates adjuvant-induced arthritis in rats. *Br J Pharmacol* **153**, 992-1002 (2008).

Even with encouraging preclinical animal models, it's always difficult to know exactly why a company decides not to develop a potential drug candidate as a therapeutic for humans. Issues related to the drug per se, such as pharmacodynamics and toxicity, as well as financial issues internal to the company and external to the market place can play important roles in that decision. In addition, while preclinical animal models are extremely useful, their pathogenesis may differ in important ways from the human disease. Thus, it is also possible that a given molecular pathway and target might be critical for a preclinical model but dispensable for the human disease. Furthermore, previous studies that targeted the CXCR1/CXCR2 chemokine system have targeted the receptors, CXCR1 and CXCR2, or a single CXCR1/CXCR2 chemokine ligand. In contrast, our study is novel in that it targeted multiple CXCR2 chemokine ligands. It is possible that targeting multiple chemokine ligands might be more efficacious than targeting the chemokine receptors. As discussed by Schall and Proudfoot⁷, effective chemokine receptor blockade might require 100% blocking of the receptor, 100% of the time. Many small molecule chemokine receptor antagonists have some agonist activity and many induce receptor internalization and recycling, making it difficult for small molecules to effectively inhibit these receptors 100% of the time. In contrast, the inherent difficulty in blocking chemokine ligands, is that there are often multiple ligands for the same receptor, limiting the effectiveness of blocking just one chemokine ligand. This is

the novelty and potential advance of our study in that we demonstrate the effectiveness of blocking multiple chemokine ligands for the same receptor.

2. In general, targeted therapies with broader specificity may have significantly more side effects. Did the authors observe any complications in the animals? Would they propose more side effects when using SA138 compared to SA129 due to the fact that CXCL2, CXCL5 blocked by the latter but not the former may have important role in host defense?

Authors' reply: We agree with the Reviewer's general comment. However, in this case, we did not observe any side effects with either treatment. Treated mice gained weight (see **Figure Supplementary Fig. 11b**) and exhibited good body conditions. There were no morbidity or mortality in either treatment group. Moreover, no signs of abnormally enlarged spleens (splenomegaly) were detected (see figure below).

Macroscopic views of mouse spleen sizes taken at day 14 from K/BxN serum transferred mice treated i.p. every day for 14 days with high doses of SA129, SA138 fusions (0.5 mL at 2 mg/mL, 1 mg/day/mice, 50 mg/kg) compared to mice injected with PBS only (0.5 mL). Representative photographs of spleens from five mice per group ($n = 5$) are shown.

3. The ELR motif has been implicated in angiogenesis. Authors target ELR-positive CXC chemokines. So when they scored the grade of inflammation, could they also score vascularity of the synovial tissue?

Authors' reply: We thank the reviewer for pointing this out. In our experience with this model over the last 12 years, however, the vascularity score is not a commonly reported measure of inflammatory arthritis in these models. In fact, we could only find two papers that evaluated the number of vessels in the joint tissue in the K/BxN serum transfer model^{8,9}. In the Isozaki T. *et al.* study⁸, von Willebrand factor (vWF) staining was used on paraffin embedded sections, and in the Amin *et al.* study⁹ CD31 staining was used on frozen sections to evaluate vascularity in joint tissue. In our laboratory, we routinely prepare paraffin-embedded samples after decalcification to preserve the tissue architecture. Therefore, we were able to stain our joint sections with a vWF antibody to evaluate the vascularity in joint tissue in each group of mice. We found that that the number of vWF-positive vessels in the joint tissue of SA138 treated mice was markedly

lower than the other groups, suggesting that angiogenesis, in addition to inflammation, might be also suppressed by treatment with SA138. However, since inflammation can drive angiogenesis, we are circumspect in concluding that the effect on angiogenesis was a direct effect on inhibiting CXCR2 chemokine ligands. These new data are shown below and have been also included as **Supplementary Fig. 12** of the revised manuscript and commented on page 13.

Arthritogenic serum was injected into C57/BL6 on days 0 and 2. Mice were treated i.p. daily with SA129, SA138 and SA^{CTR} fusions (0.5 mL at 2 mg/mL, 1 mg/day/mice) beginning on day 0. a) Representative immunofluorescent staining of ankle tissue sections harvested on day 8 from mice treated with SA129 (red), SA138 (blue) and control SA^{CTR} (gray). Scale bar represents 50 μ m. White arrows indicate von Willebrand factor-positive (vWF⁺) vessels; b) Quantitation of vWF⁺ vessels per high-powered field (hpf) in the joint ($n = 3$ mice per condition). P values: * $P < 0.05$.

4. What was the rationale of measuring ankle thickness rather than paw volume or ankle circumference?

Authors' reply: Measuring ankle thickness with a caliper is a commonly used technique to evaluate arthritis in mouse models, including the K/BxN serum transfer model¹⁰⁻¹² and the type II collagen-induced arthritis (CIA) model¹³⁻¹⁵. This method is described in detail in a methodological paper¹⁶.

5. Authors write "We next quantified the number of synovial fluid neutrophils isolated from the arthritic joints of mice treated with SA129, SA138, and SA^{CTR} antibody fusions. Synovial tissues were harvested at the peak of the disease (day 8 after disease initiation). We discovered that..." I am a bit confused. Did they actually assess synovial fluid neutrophils or synovial tissue cells? Certainly, in a chemokine study, the quantification of synovial tissue cells would be more relevant.

Authors' reply: We quantitated the number of neutrophils in the synovial fluid as we have done in all of our prior published chemokine studies¹⁷⁻²⁰. We believe the quantification of synovial fluid neutrophils is relevant to a chemokine study. Neutrophils are the major immune cell type present in RA synovial fluid comprising 60%-90% of the leukocytes found in the RA synovial fluid²¹. For still unknown reasons, however,

neutrophils are not efficiently retained in the synovial tissue²². We believe this has led to an underappreciation of the role of neutrophils in the pathogenesis of RA. The preponderance of data however suggests that neutrophils are an important cell type that drives joint destruction in RA²². In addition to the plethora of bioactive mediators released by neutrophils into the joint, several recent studies have found that neutrophil extracellular traps (NETs) in the joint fluid are a source of citrullinated autoantigens^{23,24}. Since the generation of anti-CCP antibodies is associated with disease severity and anti-CCP antibodies are detected early in RA, we believe that the early recruitment of neutrophils into the joint is important in the pathogenesis of RA. In addition, in our recent prior study using mixed bone marrow chimeric mice, we found that CXCR2 played a unique and important role in the accumulation of neutrophils in the joint, and this was due in part to preventing neutrophil apoptosis in the joint²⁰. We therefore believe that targeting neutrophil recruitment into the joint space early in RA could be a novel therapeutic strategy.

6. Finally, taking together their data and previous failures, would the authors summarize why their approach would be more successful and would lack all issues that led to failure of previous studies?

Authors' reply: As mentioned above, previous studies that targeted the CXCR1/CXCR2 chemokine system targeted the receptors, CXCR1 and CXCR2. In contrast, our study targeted multiple CXCR2 ligands. It is possible that targeting multiple chemokine ligands might be more efficacious than targeting the chemokine receptors. As discussed by Schall and Proudfoot¹, effective chemokine receptor blockade might require 100% blocking of the receptor 100% of the time. Many small molecule chemokine receptor antagonists have some agonist activity and many induce receptor internalization and recycling making it difficult for small molecules to effectively inhibit these receptors 100% of the time. The inherent difficulty in blocking chemokine ligands, is that there are often multiple ligands for the same receptor, limiting the effectiveness of blocking just one chemokine ligand. This is the novelty and potential advance of our study in that we demonstrate the effectiveness of blocking multiple chemokine ligands for the same receptor.

We again thank the reviewer for the overall positive comments and for his/her appreciation of our work.

References cited:

1. Proudfoot, A.E., Bonvin, P. & Power, C.A (2015). Targeting chemokines: Pathogens can, why can't we? *Cytokine* 74, 259-67.

2. Felix, J. & Savvides, S.N. (2017). Mechanisms of immunomodulation by mammalian and viral decoy receptors: insights from structures. *Nat. Rev. Immunol.* 17, 112-29.
3. Counago, R.M. *et al.* (2015). Structures of Orf Virus Chemokine Binding Protein in Complex with Host Chemokines Reveal Clues to Broad Binding Specificity. *Structure* 23, 1199-213.
4. Ruiz-Argüello, M.B., Smith, V.P., Campanella, G.S.V., Baleux, F., Arenzana-Seisdedos, F., Luster, A.D., and Alcami, A. (2008). An ectromelia virus protein that interacts with chemokines through their glycosaminoglycan binding domain. *J. Virol.* 82, 917–926.
5. Antonets, D.V., Nepomnyashchikh, T.S., and Shchelkunov, S.N. (2010). SECRET domain of variola virus CrmB protein can be a member of poxviral type II chemokine-binding proteins family. *BMC Res. Notes* 3, 271.
6. Xue, X., Lu, Q., Wei, H., Wang, D., Chen, D., He, G., Huang, L., Wang, H., and Wang, X. (2011). Structural basis of chemokine sequestration by CrmD, a poxvirus-encoded tumor necrosis factor receptor. *PLoS Pathog.* 7, e1002162.
7. Schall, T.J., and Proudfoot, A.E. (2011). Overcoming hurdles in developing successful drugs targeting chemokine receptors. *Nat. Rev. Immunol.* 11, 355-63.
8. Amin, M.A., Campbell, P.L., Ruth, J.H., Isozaki, T., Rabquer, B.J., Alex Stinson, W., O'Brien, M., Edhayan, G., Ohara, R.A., Vargo, J., *et al.* (2015). A key role for Fut1-regulated angiogenesis and ICAM-1 expression in K/BxN arthritis. *Ann. Rheum. Dis.* 74, 1459-66.
9. Isozaki, T., Arbab, A.S., Haas, C.S., Amin, M.A., Arendt, M.D., Koch, A.E., and Ruth, J.H. (2013). Evidence that CXCL16 is a potent mediator of angiogenesis and is involved in endothelial progenitor cell chemotaxis : studies in mice with K/BxN serum-induced arthritis. *Arthritis Rheum.* 65, 1736-46.
10. Choe, J.Y., Crain, B., Wu, S.R., and Corr, M. (2003). Interleukin 1 receptor dependence of serum transferred arthritis can be circumvented by toll-like receptor 4 signaling. *J. Exp. Med.* 197, 537-42.
11. Cunin, P., Penke, L.R., Thon, J.N., Monach, P.A., Jones, T., Chang, M.H., Chen, M.M., Melki, I., Lacroix, S., Iwakura, Y., *et al.* (2017). Megakaryocytes compensate for Kit insufficiency in murine arthritis. *J. Clin. Invest.* 127, 1714-24.
12. Katayama, M., Ohmura, K., Yukawa, N., Terao, C., Hashimoto, M., Yoshifuji, H., Kawabata, D., Fujii, T., Iwakura, Y., and Mimori, T. (2013). Neutrophils are essential as a source of IL-17 in the effector phase of arthritis. *PLoS One* 8, e62231.

13. Fukuda, S., Kohsaka, H., Takayasu, A., Yokoyama, W., Miyabe, C., Miyabe, Y., Harigai, M., Miyasaka, N., and Nanki, T. (2014). Cannabinoid receptor 2 as a potential therapeutic target in rheumatoid arthritis. *BMC Musculoskelet. Disord.* 15, 275.
14. Paniagua, R.T., Sharpe, O., Ho, P.P., Chan, S.M., Chang, A., Higgins, J.P., Tomooka, B.H., Thomas, F.M., Song, J.J., Goodman, S.B., *et al.* (2006). Selective tyrosine kinase inhibition by imatinib mesylate for the treatment of autoimmune arthritis. *J. Clin. Invest.* 116, 2633-42.
15. Yoshioka, Y., Kozawa, E., Urakawa, H., Arai, E., Futamura, N., Zhuo, L., Kimata, K., Ishiguro, N., and Nishida, Y. (2013). Suppression of hyaluronan synthesis alleviates inflammatory responses in murine arthritis and in human rheumatoid synovial fibroblasts. *Arthritis Rheum.* 65, 1160-70.
16. Rosloniec, E.F., Cremer, M., Kang, A.H., Myers, L.K., and Brand, D.D. (2010). Collagen-induced arthritis. *Curr. Protoc. Immunol.* 89, 15.5:15.5.1–15.5.25.
17. Chou, R.C., Kim, N.D., Sadik, C.D., Seung, E., Lan, Y., Byrne, M.H., Haribabu, B., Iwakura, Y., and Luster, A.D. (2010). Lipid-cytokine-chemokine cascade drives neutrophil recruitment in a murine model of inflammatory arthritis. *Immunity* 33, 266-78.
18. Kim, N.D., Chou, R.C., Seung, E., Tager, A.M., and Luster, A.D. (2006). A unique requirement for the leukotriene B4 receptor BLT1 for neutrophil recruitment in inflammatory arthritis. *J. Exp. Med.* 203, 829-35.
19. Miyabe, Y., Kim, N.D., Miyabe, C., and Luster, A.D. (2016). Studying Chemokine Control of Neutrophil Migration In Vivo in a Murine Model of Inflammatory Arthritis. *Methods Enzymol.* 570, 207-231.
20. Miyabe, Y., Miyabe, C., Murooka, T.T., Kim, E.Y., Newton, G.A., Kim, N.D., Haribabu, B., Lusinskas, F.W., Mempel, T.R., and Luster, A.D. (2017). Complement C5a Receptor is the Key Initiator of Neutrophil Adhesion Igniting Immune Complex-induced Arthritis. *Sci. Immunol.* 2.
21. Watson, F., Robinson, J.J., Phelan, M., Bucknall, R.C., and Edwards, S.W. (1993). Receptor expression in synovial fluid neutrophils from patients with rheumatoid arthritis. *Ann. Rheum. Dis.* 52, 354-9.
22. Wright, H.L., Moots, R.J., and Edwards, S.W. (2014). The multifactorial role of neutrophils in rheumatoid arthritis. *Nat. Rev. Rheumatol.* 10, 593-601.
23. Carmona-Rivera, C., Carlucci, P.M., Moore, E., Lingampalli, N., Uchtenhagen, H., James, E., Liu, Y., Bicker, K.L., Wahamaa, H., Hoffmann, V., *et al.* (2017).

Synovial fibroblast-neutrophil interactions promote pathogenic adaptive immunity in rheumatoid arthritis. *Sci. Immunol.* 2.

24. Khandpur, R., Carmona-Rivera, C., Vivekanandan-Giri, A., Gizinski, A., Yalavarthi, S., Knight, J.S., Friday, S., Li, S., Patel, R.M., Subramanian, V., *et al.* (2013). NETs are a source of citrullinated autoantigens and stimulate inflammatory responses in rheumatoid arthritis. *Sci. Transl. Med.* 5, 178ra140.

REVIEWERS' COMMENTS:

Reviewer #1 (Remarks to the Author):

I am happy with the modifications made.

Reviewer #3 (Remarks to the Author):

I am happy with the changes. I have no further comments.